# Deep Koopman-layered Model with Universal Property Based on Toeplitz Matrices

## Abstract

We propose deep Koopman-layered models with learnable parameters in the form of Toeplitz matrices for analyzing the dynamics of time-series data. The proposed model has both theoretical solidness and flexibility. By virtue of the universal property of Toeplitz matrices and the reproducing property underlined in the model, we can show its universality and the generalization property. In addition, the flexibility of the proposed model enables the model to fit time-series data coming from nonautonomous dynamical systems. When training the model, we apply Krylov subspace methods for efficient computations. In addition, the proposed model can be regarded as a neural ODE-based model. In this sense, the proposed model establishes a new connection among Koopman operators, neural ODEs, and numerical linear algebraic methods.

## 1 Introduction

Koopman operator has been one of the important tools in machine learning (Kawahara, 2016; Ishikawa et al., 2018; Lusch et al., 2017; Brunton & Kutz, 2019; Hashimoto et al., 2020). Koopman operators are linear operators that describe the composition of functions and are applied to analyzing time-series data generated by nonlinear dynamical systems (Koopman, 1931; Budišić et al., 2012; Klus et al., 2020; Giannakis & Das, 2020; Mezić, 2022). For systems with discrete Koopman spectra, by computing the eigenvalues of Koopman operators, we can understand the long-term behavior of the undelined dynamical systems. An important feature of Applying Koopman operators is that we can estimate them with given time-series data through fundamental linear algebraic tools such as projection. A typical approach to estimate Koopman operators is extended dynamical mode decomposition (EDMD) (Williams et al., 2015). For EDMD, we need to choose the dictionary functions to determine the representation space of the Koopman operator, and what choice of them gives us a better estimation is far from trivial. In addition, since we construct the estimation in an analytical way, the model is not flexible enough to incorporate additional information about dynamical systems. With EDMD as a starting point, many DMD-based methods are proposed (Kawahara, 2016; Colbrook & Townsend, 2024; Schmid, 2022). For autonomous systems, we need to estimate a single Koopman operator. In this case, Ishikawa et al. (2024) proposed to choose derivatives of kernel functions as dictionary functions based on the theory of Jet spaces. Several works deal with nonautonomous systems. Maćešić et al. (2018) applied EDMD to estimate a time-dependent Koopman operator for each time window. Peitz & Klus (2019) applied EDMD for switching dynamical systems for solving optimal control problems. However, as far as we know, no existing works show proper choices of dictionary functions for nonautonomous systems based on theoretical analysis. In addition, in the above approaches for nonautonomous systems, since each Koopman operator for a time window is estimated individually, we cannot take the information of other Koopman operators into account.

To find a proper representation space and gain the flexibility of the model, neural network-based Koopman methods have been proposed (Lusch et al., 2017; Azencot et al., 2020; Shi & Meng, 2022). These methods set the encoder from the data space to the representation space where the Koopman operator is defined, and the decoder from the representation space to the data space, as deep neural networks. Then, we train them. Neural network-based Koopman methods for nonautonomous systems have also been proposed. Liu et al. (2023) proposed to decompose the Koopman operator into a time-invariant part and a time-variant part. The time-variant part of the Koopman operator is constructed individually for each time window using EDMD. Xiong et al. (2024) assumed the ergodicity of the dynamical system and considered time-averaged Koopman

operators for nonautonomous dynamical systems. However, their theoretical properties have not been fully understood, and since the representation space changes as the learning process proceeds, their theoretical analysis is challenging.

In this work, we propose a framework that estimates multiple Koopman operators over time with the Fourier basis representation space and learnable Toeplitz matrices. Using our framework, we can estimate multiple Koopman operators simultaneously and can capture the transition of properties of data along time via multiple Koopman operators. We call each Koopman operator the Koopman-layer, and the whole model the deep Koopman-layered model. The proposed model has both theoretical solidness and flexibility. We show that the Fourier basis is a proper basis for constructing the representation space even for nonautonomous dynamical systems in the sense that we can show its theoretical properties such as universality and generalization bound. In addition, the proposed model has learnable parameters, which makes the model more flexible to fit nonautonomous dynamical systems than the analytical methods such as EDMD. The proposed model resolves the issue of theoretical analysis for the neutral network-based methods and that of the flexibility for the analytical methods simultaneously.

We show that each Koopman operator is represented by the exponential of a matrix constructed with Toeplitz matrices and diagonal matrices. This allows us to apply Krylov subspace methods (Gallopoulos & Saad, 1992; Güttel, 2013; Hashimoto & Nodera, 2016) to compute the estimation of Koopman operators with low computational costs. By virtue of the universal property of Toeplitz matrices (Ye & Lim, 2016), we can show the universality of the proposed model with a linear algebraic approach. We also show a generalization bound of the proposed model using a reproducing kernel Hilbert space (RKHS) associated with the Fourier functions. We can analyze both the universality and generalization error with the same framework.

The proposed model can also be regarded as a neural ODE-based model (Chen et al., 2018; Teshima et al., 2020a; Li et al., 2023). While in the existing method, we train the models with numerical analysis approaches, in the proposed method, we train the models with a numerical linear algebraic approach. The universality and generalization results of the proposed model can also be seen as those for the neural ODE-based models. Our method sheds light on a new linear algebraic approach to the design of neural ODEs.

Our contributions are summarized as follows:

- We propose a model for analyzing nonautonomous dynamical systems that has both theoretical solidness and flexibility. We show that the Fourier basis provides us with a proper representation space, in the sense that we can show the universality and the generalization bound regarding the model. As for the flexibility, we can learn multiple Koopman operators simultaneously, which enables us to extract the transition of properties of dynamical systems along time.
- We apply Krylov subspace methods to compute the estimation of Koopman operators. This establishes a new connection between Koopman operator theoretic approaches and Krylov subspace methods, which opens up future directions for extracting further information about dynamical systems using numerical linear algebraic approaches.
- We provide a new implementation method for neural ODEs purely with numerical linear algebraic approaches, not with numerical analysis approaches.

## 2 PRELIMINARY

### 2.1 NOTATIONS

In this paper, we use a generalized concept of matrices. For a finite index set $N \subset \mathbb{Z}^d$ and $a_{j,l} \in \mathbb{C}$ $(j, l \in N)$, we call $A = [a_{j,l}]_{j,l \in N}$ an $N$ by $N$ matrix and denote by $\mathbb{C}^{N \times N}$ the space of all $N$ by $N$ matrices. Indeed, by constructing a bijection $I : N \to \{1, \ldots, |N|\}$ and setting $\tilde{a}_{I(j),I(l)} = a_{j,l}$, we obtain a standard matrix $[\tilde{a}_{I(j),I(l)}]_{I(j),I(l)}$ corresponding to $A$. Thus, we can deal with the generalized matrices in the same manner as the standard matrices.

### 2.2 $L^2$ SPACE AND REPRODUCING KERNEL HILBERT SPACE ON THE TORUS

We consider two function spaces, the $L^2$ space and RKHS, in this paper. Let $\mathbb{T}$ be the torus $\mathbb{R}/2\pi\mathbb{Z}$, i.e., the set of real numbers modulo $2\pi$. We denote by $L^2(\mathbb{T}^d)$ the space of complex-valued square-

integrable complex-valued functions on $\mathbb{T}^d$, equipped with the Lebesgue measure. As for the RKHS, let $\kappa : \mathbb{T}^d \times \mathbb{T}^d \to \mathbb{C}$ be a positive definite kernel, which satisfies the following two properties:

1. $\kappa(x, y) = \overline{\kappa(y, x)}$ for $x, y \in \mathbb{T}^d$,
2. $\sum_{n,m=1}^N \overline{c_n} c_m \kappa(x_n, x_m) \geq 0$ for $N \in \mathbb{N}$, $c_n \in \mathbb{C}$, $x_n \in \mathbb{T}$.

Let $\phi$ be a feature map defined as $\phi(x) = \kappa(\cdot, x)$. The RKHS $\mathcal{H}_\kappa$ is the Hilbert space spanned by $\{\phi(x) \mid x \in \mathbb{T}^d\}$. The inner product $\langle \cdot, \cdot \rangle : \mathcal{H}_\kappa \times \mathcal{H}_\kappa \to \mathbb{C}$ in $\mathcal{H}_\kappa$ is defined as

$$\left\langle \sum_{n=1}^N c_n \phi(x_n), \sum_{m=1}^M d_m \phi(y_m) \right\rangle = \sum_{n=1}^N \sum_{m=1}^M \overline{c_n} d_m \kappa(x_n, y_m)$$

for $c_n, d_n \in \mathbb{C}$ and $x_n, y_n \in \mathbb{T}^d$. Note that by the definition of $\kappa$, $\langle \cdot, \cdot \rangle$ is well-defined and satisfies the axiom of inner products. An important property for RKHSs is the reproducing property. For $x \in \mathbb{T}^d$ and $v \in \mathcal{H}_\kappa$, we have $\langle \phi(x), v \rangle = v(x)$, which is useful for deriving a generalization bound.

### 2.3 KOOPMAN GENERATOR AND OPERATOR

Consider an ODE $\frac{dx}{dt}(t) = f(x(t))$ on $\mathbb{T}^d$. Let $g : \mathbb{R} \times \mathbb{T}^d$ be the flow of the ODE, that is, $g$ satisfies $g(0, x) = x$ and $g(s, g(t, x)) = g(s + t, x)$ for $x \in \mathbb{T}^d$. The function $g(\cdot, x)$ is the trajectory of the dynamical system starting at the initial value $x$. We assume $g$ is continuous and invertible. We also assume the Jacobian $Jg_t^{-1}$ of $g_t^{-1}$ is bounded for any $t \in \mathbb{R}$, where $g_t = g(t, \cdot)$. We define the Koopman operator $K^t$ on $L^2(\mathbb{T}^d)$ by the composition with $g(t, \cdot)$ as $K^t h(x) = h(g(t, x))$ for $h \in L^2(\mathbb{T}^d)$ and $x \in \mathbb{T}^d$. The Koopman operator is a linear operator that maps a function $h$ to a function $h(g(t, \cdot))$. Note that the Koopman operator $K^t$ is linear even if $g(t, \cdot)$ is nonlinear. Since $K^t$ depends on $t$, we can consider the family of Koopman operators $\{K^t\}_{t \in \mathbb{R}}$. For $h \in C^1(\mathbb{T}^d)$, where $C^1(\mathbb{T}^d)$ is the space of continuous differentiable functions on $\mathbb{T}^d$, define a linear operator $L$ as

$$Lh = \lim_{t \to \infty} \frac{K^t h - h}{t},$$

where the limit is by means of $L^2(\mathbb{T})$. We call $L$ the Koopman generator. We write $K^t = e^{tL}$. Note that for the function $h$ defined as $h(t, x) = K^t \tilde{h}(x)$ for $\tilde{h} \in C^1(\mathbb{T}^d)$, we have $\frac{\partial h}{\partial t} = Lh$. If $L$ is bounded, then it coincides with the standard definition $e^{tL} = \sum_{i=1}^\infty (tL)^i / i!$. If $L$ is unbounded, it can be justified by approximating $L$ by a sequence of bounded operators and considering the strong limit of the sequence of the exponential of the bounded operators (Yosida, 1980).

## 3 DEEP KOOPMAN-LAYERED MODEL

We propose deep Koopman-layered models based on the Koopman operator theory, which have both theoretical solidness and flexibility.

### 3.1 MULTIPLE DYNAMICAL SYSTEMS AND KOOPMAN GENERATORS

Consider $J$ ODEs $\frac{dx}{dt}(t) = f_j(x(t))$ on $\mathbb{T}^d$ for $j = 1, \ldots, J$. Let $g_j : \mathbb{R} \times \mathbb{T}^d$ be the flow of the $j$th ODE. For $v \in L^2(\mathbb{T}^d)$, consider the following model:

$$G(x) = v \circ g_J(t_J, \cdot) \circ \cdots \circ g_1(t_1, \cdot)(x) = v(g_J(t_J, \cdots g_1(t_1, x))). \tag{1}$$

Starting from a point $x$, it is first transformed according to the flow $g_1$, and then $g_2$, and so on. This model describes a switching dynamical system, and also is regarded as a discrete approximation of a nonautonomous dynamical system.

**Remark 3.1** *Since we are focusing on the complex-valued function space $L^2(\mathbb{T}^d)$, $G$ itself is a complex-valued function. However, we can easily extend the model to the flow $g_J(t_J, \cdot) \circ \cdots \circ g_1(t_1, \cdot)$, which is a map from $\mathbb{T}^d$ to $\mathbb{T}^d$. We can obtain a complex-valued function on $\mathbb{T}^{d+1}$ that describes a map from $\mathbb{T}^d$ to $\mathbb{T}^d$. Indeed, let $\tilde{g}_j(x, y) = [g_j(t_j, x), y]$ for $x \in \mathbb{T}^d$ and $y \in \mathbb{T}$. Let $\tilde{v}$ be a function that satisfies $\tilde{v}(x, k/d) = x_k$, where $x_k$ is the kth element of $x$, and let $G = \tilde{v} \circ \tilde{g}_J \circ \cdots \circ \tilde{g}_1$. Then, $G(\cdot, k/d)$ is the kth element of $g_J(t_J, \cdot) \circ \cdots \circ g_1(t_1, \cdot)$.*

**Remark 3.2** *The analysis in $\mathbb{T}^d$ is not restrictive. In many practical cases, we are interested in dynamics in a bounded domain $\Omega$ in $\mathbb{R}^d$. For example, dynamics in a space around a certain object (e.g., heat source). Let $B_d$ be the unit ball in $\mathbb{T}^d$. If $\Omega$ is diffeomorphic to $B_d$, then we can construct a dynamical system $\check{f}_j$ on $\mathbb{T}^d$ that satisfies $\check{f}_j(x) = \tilde{f}_j(x)$ for $x \in B_d$, where $\tilde{f}_j$ is the equivalent dynamical system on $B_d$ with $f_j$. In addition, although we focus on the fundamental case of $\mathbb{T}^d$, the analysis on $\mathbb{T}^d$ opens up methods for more general cases. Indeed, $\mathbb{T}^d$ is the simplest example of locally compact groups, and the Fourier functions are generalized to the irreducible representations (Fulton & Harris, 2004). See Appendix B for more details.*

3.2 APPROXIMATION OF KOOPMAN GENERATORS USING TOEPLITZ MATRICES

We consider training the model (1) using given time-series data. For this purpose, we apply the Koopman operator theory. Let $L_j$ be the Koopman generator associated with the flow $g_j$. Since the Koopman operator $K_j^{t_j}$ of $g_j$ is represented as $\mathrm{e}^{t_j L_j}$, the model (1) is represented as

$$G = \mathrm{e}^{t_1 L_1} \cdots \mathrm{e}^{t_J L_J} v.$$

To deal with the Koopman generators defined on the infinite-dimensional space, we approximate them using a finite number of Fourier functions. For the remaining part of this section, we omit the subscript $j$ for simplicity. However, in practice, the approximation is computed for the generator $L_j$ for each layer $j = 1, \ldots, J$. Let $q_n(x) = \mathrm{e}^{in \cdot x}$ for $n \in \mathbb{Z}^d$ and $x \in \mathbb{T}^d$, where i is the imaginary unit. Let $M_r \subset \mathbb{Z}^d$ be a finite index set for $r = 1, \ldots, R$. We set the $k$th element of the function $f$ in the ODE as

$$\sum_{m_R \in M_R} a_{m_R, R}^k q_{m_R} \cdots \sum_{m_1 \in M_1} a_{m_1, 1}^k q_{m_1} \tag{2}$$

with $a_{m_r, r}^k \in \mathbb{C}$, the product of weighted sums of Fourier functions. Then, we approximate the Koopman generator $L$ by projecting the input vector onto the finite-dimensional space $V_N := \mathrm{Span}\{q_n \mid n \in N\}$, where $N \subset \mathbb{Z}^d$ is a finite index set, applying $L$, and projecting it back to $V_N$ as $Q_N Q_N^* L Q_N Q_N^*$. Here, $Q_N : \mathbb{C}^N \to V_N$ is the linear operator defined as $Q_N c = \sum_{n \in N} c_n q_n$ for $c = (c_n)_{n \in N} \in \mathbb{C}^N$ and $*$ is the adjoint. Note that $Q_N Q_N^*$ is the projection onto $V_N$. Then, the representation matrix $Q_N^* L Q_N$ of the approximated Koopman generator $Q_N Q_N^* L Q_N Q_N^*$ is written as follows. Throughout the paper, all the proofs are documented in Appendix A.

**Proposition 3.3** *The $(n, l)$-entry of the representation matrix $Q_N^* L Q_N$ of the approximated operator is*

$$\sum_{k=1}^d \sum_{n_R - l \in M_R} \sum_{n_{R-1} - n_R \in M_{R-1}} \cdots \sum_{n_2 - n_3 \in M_2} \sum_{n - n_2 \in M_1}$$
$$a_{n_R - l, R}^k a_{n_{R-1} - n_R, R-1}^k \cdots a_{n_2 - n_3, 2}^k a_{n - n_2, 1}^k il_k, \tag{3}$$

*where $l_k$ is the $k$th element of the index $l \in \mathbb{Z}^d$. Moreover, we set $n_r = m_{R_j} + \cdots + m_r + l$, thus $n_1 = n$, $m_r = n_r - n_{r+1}$ for $r = 1, \ldots, R-1$, and $m_R = n_R - l$.*

Note that since the sum involves the differences of indices, it can be written using Toeplitz matrices, whose $(n, l)$-entry depends only on $n - l$. We approximate the sum appearing in Eq. (3) by restricting the index $n_r$ to $N$, combining with the information of time $t$, and setting a matrix $\mathbf{L} \in \mathbb{C}^{N \times N}$ as

$$\mathbf{L} = t \sum_{k=1}^d A_1^k \cdots A_R^k D_k, \tag{4}$$

where $A_r^k$ is the Toeplitz matrix defined as $A_r^k = [a_{n-l,r}^k]_{n,l \in N}$ and $D_k$ is the diagonal matrix defined as $(D_k)_{l,l} = il_k$ for the index $l \in \mathbb{Z}^d$. We finally regard $Q_N \mathbf{L} Q_N^*$ as an approximation of the Koopman genertor $L$.

Then, we construct the approximation $\mathbf{G}$ of $G$, defined in Eq. (1), as

$$\mathbf{G} = \mathrm{e}^{Q_N \mathbf{L}_1 Q_N^*} \cdots \mathrm{e}^{Q_N \mathbf{L}_J Q_N^*} v = Q_N \mathrm{e}^{\mathbf{L}_1} \cdots \mathrm{e}^{\mathbf{L}_J} Q_N^* v. \tag{5}$$

We call the model $\mathbf{G}$ deep Koopman-layered model.

To compute the product of the matrix exponential $\mathrm{e}^{\mathbf{L}_j}$ and the vector $\mathrm{e}^{\mathbf{L}_{j+1}} \cdots \mathrm{e}^{\mathbf{L}_J} Q_N^* v$, we can use Krylov subspace methods. If the number of indices for describing $f$ is smaller than that for describing the whole model, i.e., $|M_r| \ll |N|$, then the Toeplitz matrix $A_r^k$ is sparse. In this case, the matrix-vector product can be computed with the computational cost of $O(\sum_{r=1}^{R} |M_r||N|)$. Thus, one iteration of the Krylov subspace method costs $O(\sum_{r=1}^{R} |M_r||N|)$, which makes the computation efficient compared to direct methods without taking the structure of the matrix into account, whose computational cost results in $O(|N|^3)$. We also note that even if the Toeplitz matrices are dense, the computational cost of one iteration of the Krylov subspace method is $O(|N| \log |N|)$ if we use the fast Fourier transform.

**Remark 3.4** *To restrict $f$ to be a real-valued map and reduce the number of parameters $a_{m,r}^k$, we can set $M_r$ as $\{-m_{1,r}, \ldots, m_{1,r}\} \times \cdots \times \{-m_{d,r}, \ldots, m_{d,r}\}$ for $m_{k,r} \in \mathbb{N}$ for $k = 1, \ldots, d$. In addition, we set $a_{m,r}^k = \overline{a_{-m,r}^k}$ for $m \in M_r$. Then, we have $a_{m,r}^k q_m = \overline{a_{-m,r}^k q_{-m}}$, and $f$ is real-valued.*

**Remark 3.5** *An advantage of applying Koopman operators is that their spectra describe the properties of dynamical systems. For example, if the dynamical system is measure preserving, then the corresponding Koopman operator is unitary. Since each Koopman layer is an estimation of the Koopman operator, we can analyze time-series data coming from nonsutonomous dynamical systems by computing the eigenvalues of the Koopman layers. We will observe the eigenvalues of Koopman layers numerically in Subsection 6.3.*

## 4 UNIVERSALITY

In this section, we show the universal property of the proposed deep Koopman-layered model. We can interpret the model $\mathbf{G}$ as the approximation of the target function by transforming the function $v$ into the target function using the linear operator $Q_N \mathrm{e}^{\mathbf{L}_1} \cdots \mathrm{e}^{\mathbf{L}_J} Q_N^*$. If we can represent any linear operator by $\mathrm{e}^{\mathbf{L}_1} \cdots \mathrm{e}^{\mathbf{L}_J}$, then we can transform $v$ into any target function in $V_N$, which means we can approximate any function as $N$ goes to the whole set $\mathbb{Z}^d$. Thus, this property corresponds to the universality of the model. In Section 3, by constructing the model with the matrix $\mathrm{e}^{\mathbf{L}_1} \cdots \mathrm{e}^{\mathbf{L}_J}$ based on the Koopman operators with the Fourier functions, we restrict the number of parameters of the linear operator that transforms $v$ into the target function. The universality of the model means that this restriction is reasonable in the sense of representing the target functions using the deep Koopman-layered model.

Let $T(N, \mathbb{C}) = \{\sum_{k=1}^{d} A_1^k \cdots A_{R_k}^k D_k, \mid R_k \in \mathbb{N}, \ A_1^k \cdots A_{R_k}^k \in \mathbb{C}^{N \times N} : \text{Toeplitz}\}$ be the set of matrices in the form of $\mathbf{L}$ in Eq. (4). Let $L_0^2(\mathbb{T}^d) = \overline{\text{Span}\{q_n \mid n \neq 0\}}$ be the space of $L^2$ functions whose average is 0. We show the following fundamental result of the universality of the model:

**Theorem 4.1** *Assume $v \in L_0^2(\mathbb{T}^d)$ and $v \neq 0$. For any $f \in L_0^2(\mathbb{T}^d)$ with $f \neq 0$ and for any $\epsilon > 0$, there exist a finite set $N \subset \mathbb{Z} \setminus \{0\}$, a positive integer $J$, and matrices $\mathbf{L}_1, \ldots, \mathbf{L}_J \in T(N, \mathbb{C})$ such that $\|f - \mathbf{G}\| \leq \epsilon$ and $\mathbf{G} = Q_N \mathrm{e}^{\mathbf{L}_1} \cdots \mathrm{e}^{\mathbf{L}_J} Q_N^* v$.*

Theorem 4.1 is for a single function $f$, but applying Theorem 4.1 for each component of $\mathbf{G}$, we obtain the following result for the flow $g_{\tilde{J}}(t_{\tilde{J}}, \cdot) \circ \cdots \circ g_1(t_1, \cdot)$ with $\tilde{J} \in \mathbb{N}$, which is considered in Eq. (1).

**Corollary 4.2** *Assume $v \in L_0^2(\mathbb{T}^d)$ and $v \neq 0$. For any sequence $g_1(t_1, \cdot), \ldots, g_{\tilde{J}}(t_{\tilde{J}}, \cdot)$ of flows that satisfies $v \circ g_{\tilde{J}}(t_{\tilde{J}}, \cdot) \circ \cdots \circ g_j(t_j, \cdot) \in L_0^2(\mathbb{T}^d)$ and $v \circ g_{\tilde{J}}(t_{\tilde{J}}, \cdot) \circ \cdots \circ g_j(t_j, \cdot) \neq 0$ for $j = 1, \ldots \tilde{J}$, and for any $\epsilon > 0$, there exist a finite set $N \subset \mathbb{Z} \setminus \{0\}$, integers $0 < J_1 < \cdots < J_{\tilde{J}}$, and matrices $\mathbf{L}_1, \ldots, \mathbf{L}_{J_{\tilde{J}}} \in T(N, \mathbb{C})$ such that $\|v \circ g_{\tilde{J}}(t_{\tilde{J}}, \cdot) \circ \cdots \circ g_j(t_j, \cdot) - \mathbf{G}_j\| \leq \epsilon$ and $\mathbf{G}_j = Q_N \mathrm{e}^{\mathbf{L}_{J_{j-1}+1}} \cdots \mathrm{e}^{\mathbf{L}_{J_j}} Q_N^* v$ for $j = 1, \ldots, \tilde{J}$, where $J_0 = 1$.*

**Remark 4.3** *The function space $L_0^2(\mathbb{T}^d)$ for the target function is not restrictive. By adding a constant to the functions in $L_0^2(\mathbb{T}^d)$, we can represent any function in $L^2(\mathbb{T}^d)$. Thus, by adding one additional learnable parameter $c \in \mathbb{C}$ to the model $\mathbf{G}$ in Theorem 4.1 and consider the model $\mathbf{G}(x) + c$ for an input $x \in \mathbb{T}^d$, we can represent any function in $L^2(\mathbb{T}^d)$.*

**Remark 4.4** *In the same manner as Theorem 4.1, we can show that we can represent any function in $V_N = \text{Span}\{q_n \mid n \in N\}$ exactly using the deep Koopman-layered model. Thus, if the decay rate of the Fourier transform of the target function is $\alpha$, then the convergence rate with respect to $N$ is $O((1 - \alpha^2)^{-d/2})$. See Appendix C for more details.*

The proof of Theorem 4.1 is obtained by a linear algebraic approach. By virtue of setting $f_j$ as the product of weighted sums of Fourier functions as explained in Eq. (2), the approximation of the Koopman generator is composed of Toeplitz matrices. As a result, we can apply the following proposition regarding Toeplitz matrices by Ye & Lim (2016, Theorem 2).

**Proposition 4.5** *For any $B \in \mathbb{C}^{N \times N}$, there exists $R = \lfloor |N| \rfloor + 1$ Toeplitz matrices $A_1, \ldots, A_R$ such that $B = A_1 \cdots A_R$.*

We use Proposition 4.5 to show the following lemma regarding the representation with $T(N, \mathbb{C})$.

**Lemma 4.6** *Assume $N \subset \mathbb{Z}^d \setminus \{0\}$. Then, we have $\mathbb{C}^{N \times N} = T(N, \mathbb{C})$.*

Since $\mathbb{C}^{N \times N}$ is a Lie algebra and the corresponding Lie group $GL(N, \mathbb{C})$, the group of nonsingular $N$ by $N$ matrices, is connected, we have the following lemma (Hall, 2015, Corollary 3.47).

**Lemma 4.7** *We have $GL(N, \mathbb{C}) = \{e^{\mathbf{L}_1} \cdots e^{\mathbf{L}_J} \mid J \in \mathbb{N}, \mathbf{L}_1, \ldots, \mathbf{L}_J \in \mathbb{C}^{N \times N}\}$.*

We also use the following transitive property of $GL(N, \mathbb{C})$ and finally obtain Theorem 4.1.

**Lemma 4.8** *For any $\mathbf{u}, \mathbf{v} \in \mathbb{C}^N \setminus \{0\}$, there exists $A \in GL(N, \mathbb{C})$ such that $\mathbf{u} = A\mathbf{v}$.*

## 5 GENERALIZATION BOUND

We investigate the generalization property of the proposed deep Koopman-layered model in this section. Our framework with Koopman operators enables us to derive a generalization bound involving the norms of Koopman operators.

Let $\mathcal{G}_N = \{Q_N e^{\mathbf{L}_1} \cdots e^{\mathbf{L}_J} Q_N^* v \mid \mathbf{L}_1, \ldots, \mathbf{L}_J \in T(N, \mathbb{C})\}$ be the function class of deep Koopman-layered model (5). Let $\ell(\mathcal{G}_N) = \{(x, y) \mapsto \ell(f(x), y) \mid f \in \mathcal{G}_N\}$ for a function $\ell$ that is bounded by $C > 0$. Then, we have the following result of a generalization bound for the deep Koopman-layered model.

**Proposition 5.1** *Let $h \in \ell(\mathcal{G}_N)$, $x$ and $y$ be random variables, $S \in \mathbb{N}$, and $x_1, \ldots, x_S$ and $y_1, \ldots, y_S$ be i.i.d. samples drawn from the distributions of $x$ and $y$, respectively. For any $\delta > 0$, with probability at least $1 - \delta$, we have*

$$\mathrm{E}[h(x, y)] \leq \frac{1}{S} \sum_{s=1}^{S} h(x_n, y_n) + \frac{\alpha}{\sqrt{S}} \max_{j \in N} e^{\tau \|j\|_1} \sup_{\mathbf{L}_1, \ldots, \mathbf{L}_J \in T(N, \mathbb{C})} \|e^{\mathbf{L}_1}\| \cdots \|e^{\mathbf{L}_J}\| \|v\| + 3C \sqrt{\frac{\log(\delta/2)}{S}}.$$

We use the Rademacher complexity to derive Proposition 5.1. For this purpose, we regard the model (1) as a function in an RKHS. For $j \in \mathbb{Z}^d$ and $x \in \mathbb{T}^d$, let $\tilde{q}_j(x) = e^{-\tau \|j\|_1} e^{ij \cdot x}$, where $\tau > 0$ is a fixed parameter and $\|[j_1, \ldots, j_d]\|_1 = |j_1| + \cdots + |j_d|$ for $[j_1, \ldots, j_d] \in \mathbb{Z}^d$. Let $\kappa(x, y) = \sum_{j \in \mathbb{Z}^d} \overline{\tilde{q}_j(x)} \tilde{q}_j(y)$, and consider the RKHS $\mathcal{H}_\kappa$ associated with the kernel $\kappa$. Note that $\kappa$ is a positive definite kernel, and $\{\tilde{q}_j \mid j \in \mathbb{Z}^d\}$ is an orthonormal basis of $\mathcal{H}_\kappa$. Giannakis et al. (2022) and Das et al. (2021) used this kind of RKHSs for simulating dynamical systems on a quantum computer based on the Koopman operator theory and for approximating Koopman operators by a sequence of compact operators. Here, we use the RKHS $\mathcal{H}_\kappa$ for deriving a generalization bound. To regard the function $\mathbf{G} \in V_N = \text{Span}\{q_j \mid j \in N\} \subset L^2(\mathbb{T}^d)$ as a function in $\mathcal{H}_\kappa$, we define an inclusion map $\iota_N : V_N \to \mathcal{H}_\kappa$ as $\iota_N q_j = e^{\tau \|j\|_1} \tilde{q}_j$ for $j \in N$. Then, the operator norm of $\iota_N$ is $\|\tau_N\| = \max_{j \in N} e^{\tau \|j\|_1}$.

Let $S \in \mathbb{N}$ be the sample size, $\sigma_1, \ldots, \sigma_S$ be i.i.d. Rademacher variables (i.e., random variables that follow uniform distribution over $\{\pm 1\}$), and $x_1, \ldots, x_S$ be given samples. Then, the empirical Rademacher complexity $\hat{R}_S(\mathcal{G}_N)$ is bounded as follows.

**Lemma 5.2** *We have*

$$\hat{R}_S(\mathcal{G}_N) \leq \frac{\alpha}{\sqrt{S}} \max_{j \in N} e^{\tau \|j\|_1} \sup_{\mathbf{L}_1, \ldots, \mathbf{L}_J \in T(N, \mathbb{C})} \|e^{\mathbf{L}_1}\| \cdots \|e^{\mathbf{L}_J}\| \|v\|,$$

*where* $\alpha = \sum_{j \in \mathbb{Z}^d} e^{-2\tau \|j\|_1}$.

We can see that the complexity of the model depends exponentially on both $N$ and $J$. Combining Lemma 4.2 in Mohri et al. (2012) and Lemma 5.2, we can derive Proposition 5.1.

**Remark 5.3** *The exponential dependence of the generalization bound on the number of layers is also typical for standard neural networks (Neyshabur et al., 2015; Bartlett et al., 2017; Golowich et al., 2018; Hashimoto et al., 2024).*

**Remark 5.4** *Based on Proposition 5.1, we can control the generalization error by adding a regularization term to the loss function to make* $\|e^{\mathbf{L}_1}\| \cdots \|e^{\mathbf{L}_J}\|$ *smaller. We note that* $\|e^{\mathbf{L}_j}\|$ *is expected to be bounded with respect to $N$ since the corresponding Koopman operator is bounded in our setting. See Appendix H for more details.*

## 6 NUMERICAL RESULTS AND PRACTICAL IMPLEMENTATION

We empirically confirm the fundamental properties of the proposed deep Koopman-layered model.

### 6.1 TRAINING DEEP KOOPMAN-LAYERED MODEL WITH TIME-SERIES DATA

Based on Corollary 4.2, we train the deep Koopman-layered model using time-series data as follows: We first fix the final nonlinear transform $v$ in the model $\mathbf{G}$ taking Remark 3.1 into account, the number of layers $\tilde{J}$, and the index sets $N$, $M_r$. We input a family of time-series data $\{x_{s,0}, \ldots, x_{s,\tilde{J}}\}_{s=1}^S$ to $\mathbf{G}$. For obtaining the output of $\mathbf{G}$, we first compute $Q_N^* v = [\langle q_n, v \rangle]_n$, where $\langle \cdot, \cdot \rangle$ is the inner product in $L^2(\mathbb{T}^d)$, and compute $e^{\mathbf{L}_J} Q_N^* v$ using the Krylov subspace method, where $J = J_{\tilde{j}}$, $\mathbf{L}_J = t_J \sum_{k=1}^d A_1^k \cdots A_R^k D_k$, and $A_r^k = [a_{n-l,r}^{k,J}]_{n,l}$ is the Toeplitz matrix. In the same manner, we compute $e^{\mathbf{L}_{J-1}}(e^{\mathbf{L}_J} Q_N^* v)$. We continue that and finally obtain the output $\mathbf{G}(x) = Q_N u(x) = \sum_{n \in N} q_n(x) u_n$, where $u = [u_1, \ldots, u_n]^T = e^{\mathbf{L}_1} \cdots e^{\mathbf{L}_J} Q_N^* v$. We learn the parameter $a_{m,r}^k$ for each layer in $\mathbf{G}$ by minimizing $\sum_{s=1}^S \ell(v(x_{s,\tilde{j}}), \mathbf{G}_j(x_{s,j-1}))$ for $j = 1, \ldots, \tilde{J}$ using an optimization method. For example, we can set an objective function $\sum_{j=1}^{\tilde{J}} \sum_{s=1}^S \ell(v(x_{s,\tilde{j}}), \mathbf{G}_j(x_{s,j-1}))$. Here $\ell : \mathbb{C} \times \mathbb{C} \to \mathbb{R}$ is a loss function. For example, we can set $\ell$ as the squared error. We documented the pseudoscope of the proposed algorithm in Appendix D.

### 6.2 REPRESENTATION POWER AND GENERALIZATION

To confirm the fundamental property of the Koopman layer, we first consider an autonomous system. Consider the van der Pol oscillator on $\mathbb{T}$

$$\frac{d^2 x(t)}{dt^2} = -\mu(1 - x(t)^2)\frac{dx(t)}{dt} + x(t), \tag{6}$$

where $\mu = 3$. By setting $dx/dt$ as a new variable, we regard Eq. (6) as a first-ordered system on the two-dimensional space. We discretized Eq. (6) with the time-interval $\Delta t = 0.01$, and generated 1000 time-series $\{x_{s,0}, \ldots x_{s,100}\}$ for $s = 1, \ldots, 1000$ with different initial values distributed uniformly on $[-1, 1] \times [-1, 1]$. We added a random noise, which was drawn from the normal distribution of mean 0 and standard deviation 0.01, to each $x_{s,j}$ and set it as $\tilde{x}_{s,j}$. For training, we used the pairs $\{x_{s,0}, \tilde{x}_{s,100}\}$ for $s = 1, \ldots, 1000$. Then, we trained deep Koopman-layered models on $\mathbb{T}^3$ by minimizing the loss $\sum_{s=1}^{1000} \|Q_N e^{\mathbf{L}_1} \cdots e^{\mathbf{L}_J} Q_N^* v(\tilde{x}_{s,0}) - \tilde{x}_{s,100}\|^2$ using the Adam optimizer (Kingma & Ba, 2015) with the learning rate 0.001. We created data for testing in the same manner as the training dataset. We set $v(x, y) = \sin(y)x_1 + \cos(y)x_2$ for $x = [x_1, x_2] \in \mathbb{T}^2$ and $y \in \mathbb{T}$. Note that based on Remark 3.1, we constructed Kooman-layers on $\mathbb{T}^{d+1}$ for the input dimension $d$, and we designed the function $v$ so that it recovers $x_1$ by $v(x, \pi/2)$ and $x_2$ by $v(x, 0)$. We used the sine and cosine functions for designing $v$ since the representation space is constructed with the Fourier

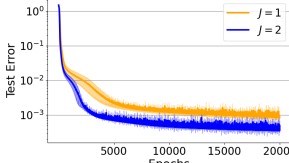 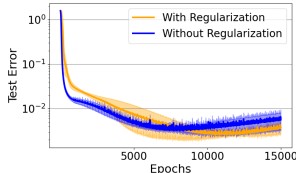

(a) Without the regularization  (b) With and without the regularization

Figure 1: Test errors for different values of $J$ with and without the regularization based on the norms of the Koopman operators. The result is the average $\pm$ the standard deviation of three independent runs.

functions. We set $N = \{n = [n_1, n_2, n_3] \in \mathbb{Z}^3 \mid -5 \leq n_1, n_2, n_3 \leq 5\} \setminus \{0\}$, $R = 1$, and $M_1 = \{n = [n_1, n_2, n_3] \in \mathbb{Z}^3 \mid -2 \leq n_1, n_2 \leq 2, -1 \leq n_3 \leq 1\} \setminus \{0\}$ for all the layers. We applied the Arnoldi method (Gallopoulos & Saad, 1992) to compute the exponential of $\mathbf{L}_j$.

Figure 1 (a) shows the test error for $J = 1$ and $J = 2$. We can see that the performance becomes higher when $J = 2$ than $J = 1$. Note that Theorem 4.1 is a fundamental result for autonomous systems, and according to Theorem 4.1, we may need more than one layer even for the autonomous systems. The result reflects this theoretical result. This is an effect of the approximation of the generator. If we can use the true Koopman generator, then we only need one layer for autonomous systems. However, since we approximated the generator using matrices, we may need more than one layer. In addition, based on Remark 5.4, we added the regularization term $10^{-5}(\|e^{\mathbf{L}_1}\| + \cdots + \|e^{\mathbf{L}_J}\|)$ and observed the behavior. We consider the case where the training data is noisy, and its sample size is small. We generated training data as above, but the sample size was 30, and the standard deviation of the noise was 0.03. We used the test data without the noise. The sample size of the test data was 1000. We set $J = 3$ to consider the case where the number of parameters is large. The result is illustrated in Figure 1 (b). We can see that with the regularization, we can achieve smaller test errors than without the regularization, which implies with the regularization, the model generalizes well.

### 6.3 EIGENVALUES OF THE KOOPMAN-LAYERS FOR NONAUTONOMOUS SYSTEMS

To confirm that we can extract information about the underlined nonautonomous dynamical systems of time-series data using the deep Koopman-layered model, we observed the eigenvalues of the Koopman-layers.

#### 6.3.1 MEASURE-PRESERVING DYNAMICAL SYSTEM

Consider the nonautonomous dynamical system on $\mathbb{T}^2$

$$\left(\frac{\mathrm{d}x_1(t)}{\mathrm{d}t}, \frac{\mathrm{d}x_2(t)}{\mathrm{d}t}\right) = \left(-\frac{\partial \zeta}{\partial x_2}(t, x(t)), \frac{\partial \zeta}{\partial x_1}(t, x(t))\right) =: f(t, x), \tag{7}$$

where $\zeta(t, [x_1, x_2]) = e^{\kappa(\cos(x_1 - t) + \cos x_2)}$. Since the dynamical system $f(t, \cdot)$ is measure-preserving for any $t \in \mathbb{R}$, the corresponding Koopman operator $K^t$ is unitary for any $t \in \mathbb{R}$. Thus, the spectrum of $K^t$ is on the unit disk in the complex plane. We discretized Eq. (7) with the time-interval $\Delta t = 0.01$, and generated 1000 time-series $\{x_{s,0}, \ldots x_{s,119}\}$ for $s = 1, \ldots, 1000$ for training with different initial values distributed uniformly on $[-1, 1] \times [-1, 1]$. We split the data into 6 subsets $S_t = \{x_{s,j} \mid s \in \{1, \ldots, 1000\}, j \in \{20t, \ldots, 20(t+1) - 1\}\}$ for $t = 0, \ldots, 5$. Then, we trained the model with 5 Kooman-layers on $\mathbb{T}^3$ by minimizing the loss $\sum_{j=1}^{5} \sum_{s=1}^{1000} \sum_{l=0}^{19} \|Q_N e^{\mathbf{L}_j} \cdots e^{\mathbf{L}_5} Q_N^* v(x_{s,20(j-1)+l}) - x_{s,100+l}\|^2$ using the Adam optimizer with the learning rate 0.001. In the same manner as Subsection 6.2, we set $v(x, y) = \sin(y)x_1 + \cos(y)x_2$ for $x = [x_1, x_2] \in \mathbb{T}^2$ and $y \in \mathbb{T}$. Note that we trained the model so that $Q_N e^{\mathbf{L}_j} \cdots e^{\mathbf{L}_5} Q_N^* v$ maps samples in $S_{j-1}$ to $S_5$. We set $N = \{n = [n_1, n_2, n_3] \in \mathbb{Z}^3 \mid -5 \leq n_1, n_2 \leq 5, -2 \leq n_3 \leq 2\}$, $R = 1$, and $M_1 = \{n = [n_1, n_2, n_3] \in \mathbb{Z}^3 \mid -2 \leq n_1, n_2 \leq 2, -1 \leq n_3 \leq 1\}$ for all the layers. We applied the Arnoldi method to compute the exponential of $\mathbf{L}_j$. In addition, we assumed the continuity of the flow of the nonautonomous dynamical system and added a regularization term $0.01 \sum_{j=2}^{5} \|e^{\mathbf{L}_j} - e^{\mathbf{L}_{j-1}}\|$ to make the Koopman layers next to each other become close. After training the model sufficiently (after 3000 epochs), we computed the eigenvalues of the approximation $e^{\mathbf{L}_j}$ of the Koopman operator for each layer $j = 1, \ldots, 5$. For comparison, we estimated the Koopman operator $K_j^{t_j}$ using EDMD and KDMD (Kawahara, 2016) with the dataset $S_{j-1}$ and $S_j$ separately

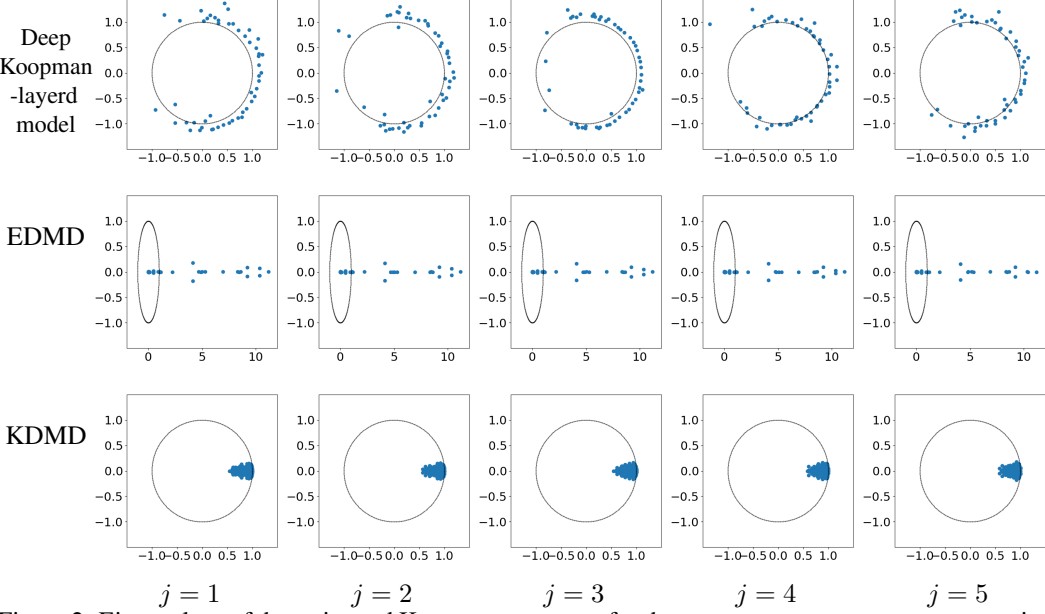

$$j = 1 \qquad j = 2 \qquad j = 3 \qquad j = 4 \qquad j = 5$$

Figure 2: Eigenvalues of the estimated Koopman operators for the nonautonomous measure preserving system.

for $j = 1, \ldots, 5$. For EDMD, we used the same Fourier functions $\{q_j \mid j \in N\}$ as the deep Koopman-layered model for the dictionary functions. For KDMD, we transformed $[x_1, x_2] \in \mathbb{T}^2$ into $\tilde{x} = [\mathrm{e}^{\mathrm{i}x_1}, \mathrm{e}^{\mathrm{i}x_2}] \in \mathbb{C}^2$ and applied the Gaussian kernel $k(x, y) = \mathrm{e}^{-0.1\|\tilde{x}-\tilde{y}\|^2}$. For estimating $K_j^{t_j}$, we applied the principal component analysis to the space spanned by $\{k(\cdot, x) \mid x \in S_{j-1}\}$ to obtain $|N|$ principal vectors $p_1, \ldots, p_{|N|}$. We estimated $K_j^{t_j}$ by constructing the projection onto the space spanned by $p_1, \ldots, p_{|N|}$. Figure 2 illustrates the results. We can see that the eigenvalues of the estimated Koopman operators by the deep Koopman-layered model are distributed on the unit circle for $j = 1, \ldots, 5$, which enables us to observe that the dynamical system is measure-preserving for any time. On the other hand, the eigenvalues of the estimated Koopman operators with EDMD and KDMD are not on the unit circle, which implies that the separately applying EDMD and KDMD failed to capture the property of the dynamical system since the system is nonautonomous.

### 6.3.2 DAMPING OSCILLATOR WITH EXTERNAL FORCE

Consider the nonautonomous dynamical system regarding a damping oscillator on a compact subspace of $\mathbb{R}$

$$\frac{\mathrm{d}^2 x(t)}{\mathrm{d}t^2} = -\alpha \frac{\mathrm{d}x(t)}{\mathrm{d}t} - x(t) - a\sin(bt), \tag{8}$$

where $\alpha = 0.1$, $a = b = 1$. By setting $\mathrm{d}x/\mathrm{d}t$ as a new variable, we regard Eq. (8) as a first-ordered system on the two-dimensional space. We generated data, constructed the deep Koopman-layered model, and applied EDMD and KDMD for comparison in the same manner as Subsection 6.3.1. Figure 3 illustrates the results. In this case, since the dynamical system is not measure preserving, it is reasonable that the estimated Koopman operators have eigenvalues inside the unit circle. We can see that many eigenvalues for the deep Koopman-layered model are distributed inside the unit circle, and the distribution changes along the layers. Since the external force becomes large as $t$ becomes large, the damping effect becomes small as $t$ becomes large (corresponding to $j$ becoming large). Thus, the number of eigenvalues distributed inside the unit circle becomes small as $j$ becomes large. On the other hand, we cannot obtain this type of observation from the separate estimation of the Koopman operators by EDMD and KDMD. See Appendix E for additional numerical results.

## 7 CONNECTION WITH OTHER METHODS

### 7.1 DEEP KOOPMAN-LAYERED MODEL AS A NEURAL ODE-BASED MODEL

The model (1) can also be regarded as a model with multiple neural ODEs (Teshima et al., 2020b; Li et al., 2023, Section 3.3). From this perspective, we can also apply the model to standard tasks with

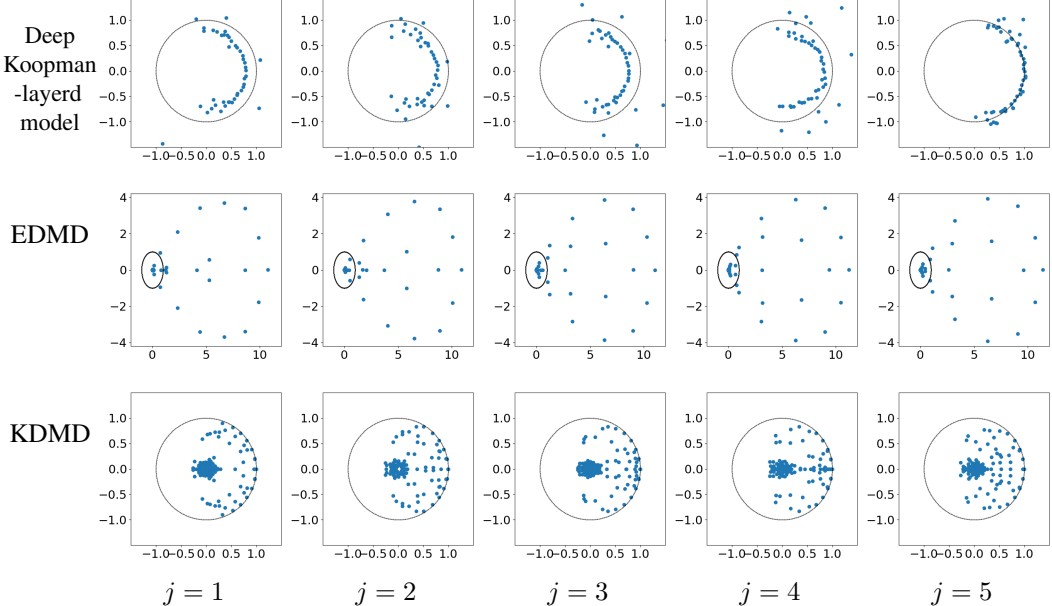

Figure 3: Eigenvalues of the estimated Koopman operators for the nonautonomous damping oscillator.

**ResNet.** For existing neural ODE-based models, we solve ODEs for the forward computation and solve adjoint equations for backward computation (Chen et al., 2018; Aleksei Sholokhov & Nabi, 2023). In our framework, solving the ODE corresponds to computing $e^{\mathbf{L}_j} u$ for a matrix $\mathbf{L}_j$ and a vector $u$. As we stated in Subsection 3.2, we use Krylov subspace methods to compute $e^{\mathbf{L}_j} u$. In this sense, our framework provides numerical linear algebraic way to solve neural ODE-based models by virtue of introducing Koopman generators and operators.

### 7.2 CONNECTION WITH NEURAL NETWORK-BASED KOOPMAN APPROACHES

In the framework of neural network-based Koopman approaches, we train an encoder $\phi$ and a decoder $\psi$ that minimizes $\|x_{t+1} - \psi(K\phi(x_t))\|$ for the given time-series $x_0, x_1, \ldots$ (Lusch et al., 2017; Li et al., 2017; Azencot et al., 2020; Shi & Meng, 2022). Here, $K$ is a linear operator, and we can construct $K$ using EDMD or can train $K$ simultaneously with $\phi$ and $\psi$. Physics-informed framework of neural network-based Koopman approaches for incorporating the knowledge of dynamics have also been proposed (Liu et al., 2024). For neural network-based Koopman approaches, since the encoder $\phi$ changes along the learning process, the representation space of the operator $K$ also changes. Thus, the theoretical analysis of these approaches is challenging. On the other hand, our deep Koopman-layered approach fixes the representation space using the Fourier functions and learns only the linear operators corresponding to Koopman generators by restricting the linear operator to a form based on the Koopman operator.

## 8 CONCLUSION AND DISCUSSION

In this paper, we proposed deep Koopman-layered models based on the Koopman operator theory combined with Fourier functions and Toeplitz matrices. We showed that the Fourier basis forms a proper representation space of the Koopman operators in the sense of the universal and generalization property of the model. In addition to the theoretical solidness, the flexibility of the proposed model allows us to train the model to fit time-series data coming from nonautonomous dynamical systems.

According to Lemma 4.7 and Theorem 4.1, to represent any function, we need more than one Koopman layer. Investigating how many layers we need and how the representation power grows as the number of layers increases theoretically remains for future work. In addition, we applied Krylov subspace methods to approximate the actions of the Koopman operators to vectors. Since the Krylov subspace methods are iterative methods, we can control the accuracy of the approximation by controlling the iteration number. How to decide and change the iteration number throughout the learning process for more efficient computations is also future work.

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

APPENDIX

## A  PROOFS

We provide the proofs of statements in the main text.

**Proposition 3.3**  *The $(n, l)$-entry of the representation matrix $Q_N^* L_j Q_N$ of the approximated operator is*

$$\sum_{k=1}^{d} \sum_{n_{R_j} - l \in M_{R_j}^j} \sum_{n_{R_j-1} - n_{R_j} \in M_{R_j-1}^j} \cdots \sum_{n_2 - n_3 \in M_2^j} \sum_{n - n_2 \in M_1^j}$$
$$a_{n_{R_j}-l, R_j}^{j,k} a_{n_{R_j-1}-n_{R_j}, R_j-1}^{j,k} \cdots a_{n_2-n_3, 2}^{j,k} a_{n-n_2, 1}^{j,k} \mathrm{i} l_k,$$

*where $l_k$ is the $k$th element of the index $l \in \mathbb{Z}^d$. Moreover, we set $n_r = m_{R_j} + \cdots + m_r + l$, thus $n_1 = n$, $m_r = n_r - n_{r+1}$ for $r = 1, \ldots, R_j - 1$, and $m_{R_j} = n_{R_j} - l$.*

**Proof**  We have

$$\langle q_n, L_j q_l \rangle = \left\langle q_n, \sum_{k=1}^{d} \sum_{m_{R_j} \in M_{R_j}^j} a_{m_{R_j}, R_j}^{j,k} q_{m_{R_j}} \cdots \sum_{m_1 \in M_1^j} a_{m_1, 1}^{j,k} q_{m_1} \mathrm{i} l_k q_l \right\rangle$$

$$= \left\langle q_n, \sum_{k=1}^{d} \sum_{m_{R_j} \in M_{R_j}^j} \cdots \sum_{m_1 \in M_1^j} a_{m_{R_j}, R_j}^{j,k} \cdots a_{m_1, 1}^{j,k} q_{m_{R_j} + \cdots + m_1 + l} \mathrm{i} l_k \right\rangle$$

$$= \sum_{k=1}^{d} \sum_{\substack{m_{R_j} + \cdots + m_1 + l = n \\ m_{R_j} \in M_{R_j}^j \cdots m_1 \in M_1^j}} a_{m_{R_j}, R_j}^{j,k} \cdots a_{m_1, 1}^{j,k} \mathrm{i} l_k$$

$$= \sum_{k=1}^{d} \sum_{n_{R_j} - l \in M_{R_j}^j} \sum_{n_{R_j-1} - n_{R_j} \in M_{R_j-1}^j} \cdots \sum_{n_2 - n_3 \in M_2^j} \sum_{n - n_2 \in M_1^j}$$
$$a_{n_{R_j}-l, R_j}^{j,k} a_{n_{R_j-1}-n_{R_j}, R_j-1}^{j,k} \cdots a_{n_2-n_3, 2}^{j,k} a_{n-n_2, 1}^{j,k} \mathrm{i} l_k.$$

$\square$

**Corollary 4.2**  *Assume $v \in L_0^2(\mathbb{T}^d)$ and $v \neq 0$. For any sequence $g_1(t_1, \cdot), \ldots, g_{\tilde{J}}(t_{\tilde{J}}, \cdot)$ of flows that satisfies $v \circ g_{\tilde{J}}(t_{\tilde{J}}, \cdot) \circ \cdots \circ g_j(t_j, \cdot) \in L_0^2(\mathbb{T}^d)$ and $v \circ g_{\tilde{J}}(t_{\tilde{J}}, \cdot) \circ \cdots \circ g_j(t_j, \cdot) \neq 0$ for $j = 1 \ldots \tilde{J}$, and for any $\epsilon > 0$, there exist a finite set $N \subset \mathbb{Z} \setminus \{0\}$, integers $0 < J_1 < \cdots < J_{\tilde{J}}$, and matrices $\mathbf{L}_1, \ldots, \mathbf{L}_{J_{\tilde{J}}} \in T(N, \mathbb{C})$ such that $\|v \circ g_{\tilde{J}}(t_{\tilde{J}}, \cdot) \circ \cdots \circ g_j(t_j, \cdot) - \mathbf{G}_j\| \leq \epsilon$ and $\mathbf{G}_j = Q_N e^{\mathbf{L}_{J_{j-1}+1}} \cdots e^{\mathbf{L}_{J_j}} Q_N^* v$ for $j = 1, \ldots, \tilde{J}$, where $J_0 = 1$.*

**Proof**  Since $v \circ g_{\tilde{J}}(t_{\tilde{J}}, \cdot) \circ \cdots \circ g_j(t_j, \cdot) \in L_0^2(\mathbb{T}^d)$ and $v \circ g_{\tilde{J}}(t_{\tilde{J}}, \cdot) \circ \cdots \circ g_j(t_j, \cdot) \neq 0$, there exist finite $N_j \subset \mathbb{Z}^d \setminus \{0\}$ and $\mathbf{G}_j \in V_{N_j}$, $\mathbf{G}_j \neq 0$ such that $\|v \circ g_{\tilde{J}}(t_{\tilde{J}}, \cdot) \circ \cdots \circ g_j(t_j, \cdot) - \mathbf{G}_j\| \leq \epsilon$ for $j = 1, \ldots, \tilde{J}$. Since $v \in L_0^2(\mathbb{T}^d)$ and $v \neq 0$, there exist finite $N_{\tilde{J}+1} \subset \mathbb{Z}^d \setminus \{0\}$ such that $Q_{N_{\tilde{J}+1}}^* v \neq 0$. Let $N = \bigcup_{j=1}^{\tilde{J}+1} N_j$. By Lemma 4.8, since $Q_N^* v \neq 0$, there exist $J_{\tilde{J}-1}, J_{\tilde{J}} \in \mathbb{N}$ and $\mathbf{L}_{J_{\tilde{J}-1}+1}, \ldots, \mathbf{L}_{J_{\tilde{J}}} \in T(N, \mathbb{C})$ such that $\mathbf{G}_{\tilde{J}} = Q_N e^{\mathbf{L}_{J_{\tilde{J}-1}+1}} \cdots e^{\mathbf{L}_{J_{\tilde{J}}}} Q_N^* v$. Since $\mathbf{G}_{\tilde{J}} \neq 0$, again by Lemma 4.8, there exist $J_{\tilde{J}-2} \in \mathbb{N}$ and $\mathbf{L}_{J_{\tilde{J}-2}+1}, \ldots, \mathbf{L}_{J_{\tilde{J}-1}} \in T(N, \mathbb{C})$ such that $\mathbf{G}_{\tilde{J}-1} = Q_N e^{\mathbf{L}_{J_{\tilde{J}-2}+1}} \cdots e^{\mathbf{L}_{J_{\tilde{J}-1}}} e^{\mathbf{L}_{J_{\tilde{J}-1}+1}} \cdots e^{\mathbf{L}_{J_1}} Q_N^* v = Q_N e^{\mathbf{L}_{J_{\tilde{J}-2}+1}} \cdots e^{\mathbf{L}_{J_{\tilde{J}-1}}} Q_N^* \mathbf{G}_{\tilde{J}}$. We continue to apply Lemma 4.8 to obtain the result. $\square$

**Lemma 4.6** *Assume $N \subset \mathbb{Z}^d \setminus \{0\}$. Then, we have $\mathbb{C}^{N \times N} = T(N, \mathbb{C})$.*

**Proof** We show $\mathbb{C}^{N \times N} \subseteq T(N, \mathbb{C})$. The inclusion $\mathbb{C}^{N \times N} \supseteq T(N, \mathbb{C})$ is trivial. Since $N \subset \mathbb{Z}^d \setminus \{0\}$, for any $n = [n_1, \ldots, n_d] \in N$, there exists $k \in \{1, \ldots, d\}$ such that $\mathrm{i} n_k = (D_k)_{n,n} \neq 0$. We denote by $k_{\min}(n)$ the minimal index $k \in \{1, \ldots, d\}$ that satisfies $(D_k)_{n,n} \neq 0$. Let $B \in \mathbb{C}^{N \times N}$. We decompose $B$ as $B = B_1 + \ldots + B_d$, where $(B_k)_{:,n} = B_{:,n}$ if $k = k_{\min}(n)$ and $(B_k)_{:,n} = \mathbf{0}$ otherwise. Here, $(B_k)_{:,n}$ is the $n$th column of $B_k$. Then, we have $(B_k)_{:,n} = \mathbf{0}$ if $(D_k)_{n,n} = 0$. Let $D_k^+$ be the diagonal matrix defined as $(D_k^+)_{n,n} = 1/(D_k)_{n,n}$ if $(D_k)_{n,n} \neq 0$ and $(D_k^+)_{n,n} = 0$ if $(D_k)_{n,n} = 0$. In addition, let $C_k = B_k D_k^+$. Then, we have $B = \sum_{k=1}^d C_k D_k$. Applying Propostion 4.5, we have $B \in T(N, \mathbb{C})$, and obtain $\mathbb{C}^{N \times N} \subseteq T(N, \mathbb{C})$. $\square$

**Lemma 4.8** *For any $\mathbf{u}, \mathbf{v} \in \mathbb{C}^N \setminus \{0\}$, there exists $A \in GL(N, \mathbb{C})$ such that $\mathbf{u} = A\mathbf{v}$.*

**Proof** Let $n_0 \in N$ and let $B \in \mathbb{N} \times \mathbb{N}$ be defined as $B_{n,:} = 1/\|\mathbf{v}\|^2 \mathbf{v}^*$ for $n = n_0$ and so that $B_{n,:}$ and $B_{m,:}$ becoming orthogonal if $n \neq m$. Then, the $n$th element of $B\mathbf{v}$ is 1 for $n = n_0$ and is 0 for $n \neq n_0$. Let $C \in \mathbb{N} \times \mathbb{N}$ be defined as $C_{n,:} = \mathbf{u}$ for $n = n_0$ and so that $C_{n,:}$ and $C_{m,:}$ becoming orthogonal if $n \neq m$. Then, $B, C \in GL(N, \mathbb{C})$ and $CB\mathbf{v} = \mathbf{u}$. $\square$

**Lemma 5.2** *We have*

$$\hat{R}_S(\mathcal{G}_N) \leq \frac{\alpha}{\sqrt{S}} \max_{j \in N} \mathrm{e}^{\tau \|j\|_1} \sup_{\mathbf{L}_1, \ldots, \mathbf{L}_J \in T(N, \mathbb{C})} \|\mathrm{e}^{\mathbf{L}_1}\| \cdots \|\mathrm{e}^{\mathbf{L}_J}\| \, \|v\|,$$

*where $\alpha = \sum_{j \in \mathbb{Z}^d} \mathrm{e}^{-2\tau \|j\|_1}$.*

**Proof**

$$\hat{R}_S(\mathcal{G}_N) = \frac{1}{S} \mathrm{E}\left[ \sup_{\mathbf{G} \in \mathcal{G}_N} \sum_{s=1}^S \mathbf{G}(x_s)\sigma_s \right] = \frac{1}{S} \mathrm{E}\left[ \sup_{\mathbf{G} \in \mathcal{G}_N} \sum_{s=1}^S \iota_N \mathbf{G}(x_s)\sigma_s \right]$$

$$= \frac{1}{S} \mathrm{E}\left[ \sup_{\mathbf{G} \in \mathcal{G}_N} \left\langle \sum_{s=1}^S \sigma_s \phi(x_s), \iota_N \mathbf{G} \right\rangle \right] \leq \frac{1}{S} \sup_{\mathbf{G} \in \mathcal{G}_N} \|\iota_N \mathbf{G}\|_{\mathcal{H}_K} \left( \sum_{s=1}^S K(x_s, x_s) \right)^{1/2}$$

$$\leq \frac{\alpha}{\sqrt{S}} \sup_{\mathbf{G} \in \mathcal{G}_N} \|\iota_N\| \|\mathbf{G}\|_{L^2(\mathbb{T}^d)} \leq \frac{\alpha}{\sqrt{S}} \max_{j \in N} \mathrm{e}^{\tau \|j\|_1} \sup_{\mathbf{L}_1, \ldots, \mathbf{L}_J \in T(N, \mathbb{C})} \|Q_N \mathrm{e}^{\mathbf{L}_1} \cdots \mathrm{e}^{\mathbf{L}_J} Q_N^* v\|$$

$$\leq \frac{\alpha}{\sqrt{S}} \max_{j \in N} \mathrm{e}^{\tau \|j\|_1} \sup_{\mathbf{L}_1, \ldots, \mathbf{L}_J \in T(N, \mathbb{C})} \|\mathrm{e}^{\mathbf{L}_1}\| \cdots \|\mathrm{e}^{\mathbf{L}_J}\| \, \|v\|,$$

where $\alpha = \sum_{j \in \mathbb{Z}^d} \mathrm{e}^{-2\tau \|j\|_1}$. $\square$

## B DETAILS OF REMARK 3.2

### B.1 REDUCTION TO THE ANALYSIS ON $\mathbb{T}^d$

If $\Omega$ is diffeomorphic to $B_d$, then we can construct a dynamical system $\check{f}_j$ on $\mathbb{T}^d$ that satisfies $\check{f}_j(x) = \tilde{f}_j(x)$ for $x \in B_d$, where $\tilde{f}_j$ is the equivalent dynamical system on $B_d$ with $f_j$. Indeed, let $B_d = \{x \in \mathbb{R}^d \mid \|x\| \leq 1\}$ be the unit ball. Let $\psi : \Omega \to B_d$ be the diffeomorphism, and let $y = \psi(x)$. Then, the dynamical system $\frac{\mathrm{d}x}{\mathrm{d}t}(t) = f_j(x(t))$ is equivalent to $\frac{\mathrm{d}y}{\mathrm{d}t}(t) = J\psi(y(t))^{-1} f_j(y(t))$ since $J\psi(y)$ is invertible for any $y \in B_d$, where $J\psi$ is the Jacobian of $\psi$. Note that since $J\psi$ does not

depend on $j$, the transition of $\tilde{f}_j$ over $j$ depends only on that of $f_j$ over $j$. Let $\tilde{f}_j(y) = J\psi(y)^{-1}f_j(y)$. Instead of considering the dynamical system $f_j$ on $\Omega$, we can consider the dynamical system $\tilde{f}_j$ on $B_d$. Let $a$ be a positive real number satisfying $1 < a < \pi$. Then, we can smoothly extend $\tilde{f}_j$ on $B_d$ to a map $\hat{f}_j$ on $aB_d$ as $\hat{f}_j(x) = \tilde{f}_j(x)$ $(x \in B_d)$, $\hat{f}_j(x) = 0$ $(\|x\| = a)$. For example, we can construct $\hat{f}_j$ in the same manner as a smooth bump function (Tu, 2011). Finally, we extend $\hat{f}_j$ on $aB_d$ to a map $\check{f}_j$ on $[-\pi, \pi]^d$ as $\check{f}_j(x) = \hat{f}_j(x)$ $(x \in aB_d)$, $\check{f}_j(x) = 0$ $(x \notin aB_d)$. Then, since $\check{f}_j([-\pi, \ldots, -\pi]) = \check{f}_j([\pi, \ldots, \pi])$, we can regard $\check{f}_j$ as a dynamical system on $\mathbb{T}^d$.

### B.2 Generalization of $\mathbb{T}^d$ to more general domain

Indeed, $\mathbb{T}^d$ is the simplest example of locally compact groups, and the Fourier functions are generalized to the irreducible representations (Fulton & Harris, 2004). For a group $G$, a representation $\rho$ is a map $\rho : G \to GL(V)$ for some vector space $V$ that satisfies $\rho(x)\rho(y) = \rho(x \cdot y)$ for $x, y \in G$. Here, $GL(V)$ is the space of all bijective linear transformations from $V$ to $V$. We note that if $G$ is abelian, then $V$ is always one-dimensional. For example, for $\mathbb{R}^d$, the irreducible representations are $\rho_\xi(x) := e^{i\xi \cdot x}$ for $x \in \mathbb{R}^d$ and $\xi \in \mathbb{R}^d$. As we will see in Section 3.2, a crucial property of the Fourier function $q_n$ is the product of two Fourier functions are also a Fourier function, i.e., $q_n \cdot q_m = q_{m+n}$. This property is valid also for general irreducible representations since we have $\rho(x)\rho(y) = \rho(x \cdot y)$. Thus, we can consider a generalized version of Toeplitz matrices by using $\rho$ (Rieffel, 2004). Although showing the universal property of the generalized Toeplitz matrices is challenging and future work, this type of argument gives us a promising way of generalizing our framework.

## C Details of Remark 4.4

In the same manner as Theorem 4.1, we can show that we can represent any function in $V_N = \text{Span}\{q_n \mid n \in N\}$ exactly using the deep Koopman-layered model. Thus, if the decay rate of the Fourier transform of the target function $h$ is $\alpha$, i.e., if there exist $0 < \alpha < 1$ such that $h$ is represented as $h = \sum_{n \in \mathbb{Z}^d} c_n q_n$ with some $c_n \in \mathbb{C}$ satisfying $|c_n| \leq \alpha^{n_1 + \cdots + n_d}$ for sufficiently large $n$, then the convergence rate with respect to $N$ is $O((1 - \alpha^2)^{-d/2})$. Indeed, for sufficiently large $N$, we have

$$\min_{\tilde{h} \in V_N} \|h - \tilde{h}\| = \left\| \sum_{n \notin N} c_n q_n \right\| = \sum_{n \notin N} |c_n|^2 \leq \sum_{n \notin N} \alpha^{2(n_1 + \cdots + n_d)} = O\left(\left(\frac{1}{1 - \alpha^2}\right)^{d/2}\right).$$

## D Algorithmic details of training deep Koopman-layered model

We provide a pseudocode of the algorithm of training the deep Koopman-layered model in Algorithm 1. Let $q_n$ be the Fourier function defined as $q_n(z) = e^{in \cdot z}$ for $n \in \mathbb{Z}^d$ and $z \in \mathbb{T}^d$, and let $\langle \cdot, \cdot \rangle$ be the inner product in $L^2(\mathbb{T}^d)$. Thus, $\langle q_n, v \rangle$ means the $n$th Fourier coefficient of a function $v$. Let $L_0^2(\mathbb{T}^d) = \overline{\text{Span}\{q_n \mid n \neq 0\}}$, and we fix a nonlinear map $v \in L_0^2(\mathbb{T}^d)$ in the model $\mathbf{G}$. We also fix the finite index set $N \subseteq \mathbb{N}^d$ determining the representation space of the Koopman generators, number of layers $J \in \mathbb{N}$, the number $R_j \in \mathbb{N}$ of Toeplitz matrices, index sets $M_1^j, \ldots, M_{R_j}^j \subseteq \mathbb{Z}^d$ determining the sparseness of the Toeplitz matrices for the $j$th layer, and the loss function $\ell : \mathbb{C} \times \mathbb{C} \to \mathbb{R}_+$. They determine the model architecture. Let $A_r^{k,j} = [a_{n-l,r}^{k,j}]_{n,l \in N, n-l \in M_r^j}$ be the Toeplitz matrix with learnable parameters $a_{n,r}^{k,j}$ and $D_k$ be the diagonal matrix with $(D_k)_{l,l} = il_k$ for $l \in \mathbb{Z}^d$. In addition, we put all the learnable parameters $A = [a_{k,j}^{n,r}]_{k=1,\ldots,d, n \in N \cap M_r^j, r=1,\ldots,R_j, j=1,\ldots\tilde{J}}$. For simplicity, we focus on the case of the number of layers $J$ is equal to the time step $\tilde{J}$. We note that the time $t$ in the definition of $\mathbf{L}$ in Subsection 3.2 do not need for practical learning algorithm since it is just regarded as the scale factor of the learnable parameter $A_1^k$.

## E Additional numerical results

---

**Algorithm 1** Training deep Koopman-layered model

---

**Require:** $v \in L_0^2(\mathbb{T})$, $N \subseteq \mathbb{Z}^d$, $J \in \mathbb{N}$, $R_1, \ldots, R_J \in \mathbb{N}$, $M_1^j, \ldots, M_{R_j}^j \subseteq \mathbb{Z}^d$ $(j = 1, \ldots, J)$,
$\quad \ell : \mathbb{C} \times \mathbb{C} \to \mathbb{R}_+$, time-series $\{x_{s,1}, \ldots, x_{s,J}\}_{s=1}^S$
**Ensure:** Learnable parameter $A$ of the deep Koopman-layered model
1: Compute a vector $u = [\langle q_n, v \rangle]_{n \in N}$.
2: Set $(D_k)_{l,l} = \mathrm{i}l_k$.
3: Initialize $A$.
4: **for** each epoch **do**
5:     **for** each layer $j = J, \ldots, 1$ **do**
6:         Compute $u = \mathrm{e}^{\sum_{k=1}^d A_1^{k,j} \cdots A_{R_j}^{k,j} D_k} u$ using a Krylov subspace method.
7:         Compute the output $y_s = \sum_{n \in N} q_n(x_{s,j-1}) u_n$ of $j$th layer for $s = 1, \ldots S$.
8:         Compute the loss $H_j = \sum_{s=1}^S \ell(v(x_{s,J}), y_s)$.
9:     **end for**
10:    Compute the total loss $H = \sum_{j=1}^J H_j$ and the gradient of $H$ with respect to $A$ and apply a
      gradient method to update the learnable parameter $A$.
11: **end for**

---

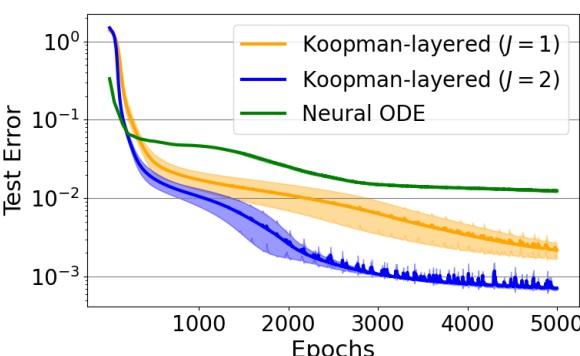

Figure 4: Test errors for the deep Koopman layered models and the neural ODE. The result is the average $\pm$ the standard deviation of three independent runs.

### E.1 COMPARISON TO NEURAL ODE

To show that the proposed model can be an alternative to neural ODE-based approaches, we conducted additional experiments. We applied a neural ODE (Chen et al., 2018) to the same problem as Subsection 6.2 (the van der Pol oscillator). The neural ODE is composed of the two-layer neural network with the hyperbolic tangent activation function whose width of the first layer is 55. The forward process is solved by the Runge-Kutta method. We note that the number of parameters of this model is $2 \times 55 + 55 \times 2 = 220$, which is almost the same as the number of parameters of the Koopman-layered model considered in Subsection 6.2, which is 222 for the case of $J = 2$. To compare the basic performance of the two models, we used one time step data for training the neural ODE. Note that in Subsection 6.2, we also used only one time step data for the deep Koopman-layered model. In the same manner as the deep Koopman-layered model, used the Adam optimizer with the learning rate 0.001. The result is shown in Figure 4. We can see that the deep Koopman-layered model outperforms the neural ODE model.

We can also use multi time step data for training the above neural ODE model. Thus, we also used two time steps $\{x_{s,0}, \tilde{x}_{s,50}, \tilde{x}_{s,100}\}_{s=1}^{1000}$ to train the same neural ODE model and compared the performance with the deep Koopman-layered model. We used the Adam optimizer with the learning rate 0.01. The result is shown in Figure 5. We can see that even if we use two time stap data for the neural ODE model, the deep Koopman-layered model with one time step data outperformed the neural ODE model. These results show that the Koopman-layered model has a potential power of being an alternative to neural ODE-based approaches.

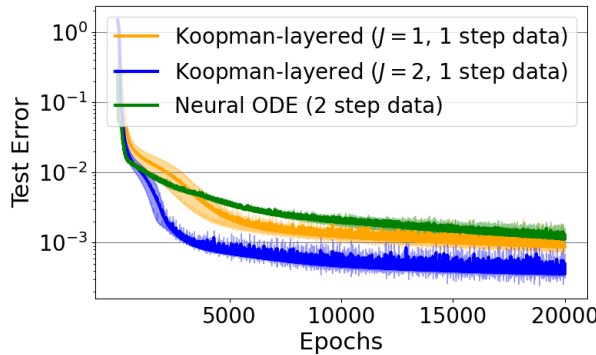

Figure 5: Test errors for the deep Koopman layered models and the neural ODE with two time step data. The result is the average $\pm$ the standard deviation of three independent runs.

### E.2 COMPARISON TO KOOPMAN-BASED APPROACH WITH LEARNED REPRESENTATION SPACES

We show the results of additional experiments with the Koopman-based approach with learned representation spaces (see Subsection 7.2). We considered the following two settings for the same example in Subsection 6.3.2.

1. Learn a set of dictionary functions to construct the representation space of five Koopman generators (learning a common set of dictionary functions is also considered by (Liu et al., 2023)).

2. Learn five sets of dictionary functions each of which is for each Koopman generator.

We used a 3-layered ReLU neural network to learn the dictionary functions. The widths of the first and the second layer are 1024 and 121. We applied the EDMD with the learned dictionary functions. The result is illustrated in Figure 6 (a,b). We cannot capture the transition of the distribution of the eigenvalues through $j = 1, \ldots, 5$ even though we learned the dictionary functions. We can also see that there are some eigenvalues equally spaced on the unit circle. This behavior is typical for autonomous systems with a constant frequency. Since the dynamical system is nonautonomous and the frequency of the system changes over time, the above behavior is not suitable for this example. This result implies that DMD-based methods try to capture the system as an autonomous system, which is not suitable for nonautonomous systems. To obtain more stable eigenvalues, we also implemented the forward-backward extended DMD (Lortie et al.) with the second setting. The result is shown in Figure 6 (c), and it is similar to the above two cases.

## F APPLICATION TO TIME-SERIES FORECASTING

We can also apply the proposed method to time-forecasting. Applying the idea of Wang et al. (2023); Liu et al. (2023), we can decompose the Koopman operators into time-invariant and time-variant parts. By extracting time-invariant features of the dynamics using the approximated Koopman operators (e.g., time-invariant eigenvectors or singular vectors), we can combine it with time-variant Koopman operators constructed by local time-series to construct the forecast. More precisely, we can decompose the Koopman operator $K^t$ for time $t$ as $K^t = K_{inv} + K_{var}^t$, where $K_{inv} = \sum_{i=1}^{n} \sigma_i v_i u_i^*$ and $K_{var} = \sum_{i=1}^{m} \tilde{\sigma}_i \tilde{v}_i \tilde{u}_i^*$, $\sigma_i, v_i, u_i$ are time-invariant singular values and the corresponding singular vectors of the approximated Koopman operators for $j = 1, \ldots, J$, $\tilde{v}_i$ are the singular vectors of the local Koopman operator that is orthogonal to $v_i$, and $\tilde{\sigma}_i$ and $\tilde{u}_i$ are singular values and singular vectors corresponding to $\tilde{v}_i$. Since we can use the time-invariant property of $t \leq t_J$, we can forecast time-series well even for $t > t_J$.

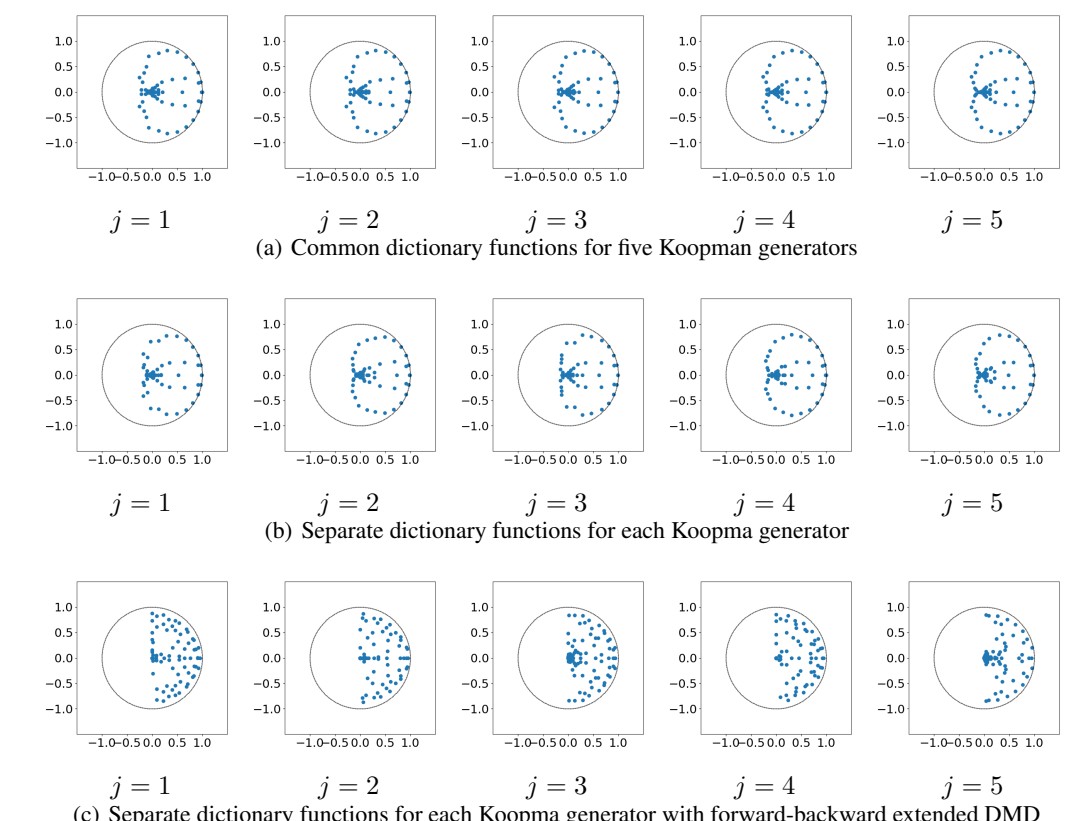

(a) Common dictionary functions for five Koopman generators

(b) Separate dictionary functions for each Koopma generator

(c) Separate dictionary functions for each Koopma generator with forward-backward extended DMD

Figure 6: Eigenvalues of the estimated Koopman operators with learned representation spaces for the nonautonomous damping oscillator.

## G  DETERMINING AN OPTIMAL NUMBER $J$ OF LAYERS

Although providing thorough discussion of determining an optimal number $J$ of layers is future work, we provide examples of heuristic approaches to determining $J$. Heuristically, we can use validation data to determine an optimal number of layers. For example, we begin by one layer and compute the validation loss. Then, we set two layers and compute the validation loss, and continue with more layers. We can set the number of layers as the number that achieves the minimal validation loss. Another way is to set a sufficiently large number of layers and train the model with the validation data. As we discussed in Section 6.2, we can add a regularization term to the loss function so that the Koopman layers next to each other become close. After the training, if there are Koopman layers next to each other and sufficiently close, then we can regard them as one Koopman layer and determine an optimal number of layers.

## H  DETAILS OF REMARK 5.4

In our setting, we assume that the flow $g(t, \cdot)$ is invertible and the Jacobian $Jg_t^{-1}$ of $g_t^{-1}$ is bounded for any $t$. Here, we denote $g_t = g(t, \cdot)$. In this case, the Koopman operator $K^t$ is bounded. Indeed, we have

$$\|K^t h\|^2 = \int_{\mathbb{T}^d} |h(g(t,x))|^2 dx = \int_{\mathbb{T}^d} |h(x)|^2 |\det Jg_t^{-1}(x)| dx \leq \|h\|^2 \sup_{x \in \mathbb{T}^d} |\det Jg_t^{-1}(x)|.$$

