# OpenReview forum: "Deep Koopman-layered Model with Universal Property Based on Toeplitz Matrices"
_ICLR.cc/2025/Conference — Submitted to ICLR 2025_

### Official Review · Reviewer_q7PE · 2024-10-23

**Soundness:** 3
**Presentation:** 2
**Contribution:** 3
**Rating:** 8
**Confidence:** 3

**Summary:**

The paper proposes a deep Koopman-layered model designed to analyze the dynamics of time-series data, particularly focusing on nonautonomous dynamical systems, where the system's behavior changes over time. The model leverages Koopman operator theory, which provides a linear framework for analyzing nonlinear dynamical systems, and it is structured as a series of "Koopman layers", each approximating a Koopman operator at a specific time point. These layers allow the model to capture the evolving dynamics of the system. Toeplitz matrices are utilized to construct the Koopman layers. Krylov subspace methods are utilized for efficient computation of the Koopman operators. The model further uses the Fourier basis functions as the representation space for the Koopman operators. This choice is justified by the theoretical properties of the Fourier basis, including its universality and the ability to derive generalization bounds for the model. The paper establishes the universality of the proposed model, meaning that it can approximate any function within the chosen representation space with arbitrary accuracy, given enough layers. Additionally, the authors derive a generalization bound, indicating the model's ability to perform well on unseen data.

**Strengths:**

The idea of using sequence of Toeplitz matrices to parameterize Koopman generator and leverage its universal property to prove the universality and generalization of the approximation scheme is novel and original. The work is solid in the sense that the framework allows explicit error control which is often missed in other Koopman operator learning literatures. Experimental results also confirms the importance of regularization as derived in the generalization bound and shows superior performance against some other approaches, in terms of the distribution of the identified eigenvalues (around the unit circle). Although the idea of “using a moving stencils method to compute the time-dependent Koopman operator (and in each local stencil, the dynamical system is assumed time invariant)” is not new (see eg. “Data-driven reduced-order modeling for nonautonomous dynamical systems in multiscale media” by Mengnan Li, Lijian Jiang), this work makes decent contribution to the workstream of Koopman operator learning for non-autonomous systems with theoretical basis.

**Weaknesses:**

- I don’t agree with the statement “For existing Neural ODE-based models, we use numerical methods, such as Runge-Kutta methods, to solve the ODE  (Chen et al., 2018)” in section 7.1. In fact, one of the innovations in that work was the authors solved the adjoint equation backwards in time to obtain the gradients automatically for parameter updates, so that one can circumvent the memory issues caused by using other numerical methods, such as Runge-Kutta, although I do agree that using Krylov subspace methods to compute $e^{L_j u}$ in the work has similar potential too.

- The notations are sometimes confusing or overloaded to digest. For example, page 3 line 161, it seems the notation of $x_k$ was not properly defined before its appearance? The superscripts and subscripts in equation (2), (3) are heavy, the authors may consider simplifying it with a bit abuse of notation OR show how it works 1D case and with 1 system so that j=1 and k=1 OR provide a more concrete examples to guide the readers through in appendix.

- In addition, consider citing two papers from Mitsubishi lab
  - "Physics-Informed Koopman Network"
  - "Physics-informed neural ODE (PINODE): embedding physics into models using collocation points"

The spirit of this work aligns quite well with above two in that both proposes to learn the Koopman generator $L$. The difference lies in that this work is “data-driven” whereas the other two are “physics/equations-driven”.

**Questions:**

- Page 7, line 358 - 359: "according to Theorem 4.1, we may need more than one layer even for the autonomous systems." -- Indeed, the theorem asserts that "there exists some positive integer $J$", but it doesn't align w/ our understanding for autonomous systems - for which the Koopman operator (or generator) should be time-invariant as defined in section 2.3? Is the statement true? How do you explain the discrepancy?

- Even for non-autonomous systems, do you have suggestions on how to choose $J$ (page 8, line 397-398) and the function $v$ beforehand? Any general guidelines? I can see because we construct the representation space  with the Fourier functions, so we could use sinusoidal functions for representing $v$ for simplicity, etc. But are there deeper reasons for why you choose $v(x,y)$ that way (page 7, line 350)?

- Page 8, 416-417: what's the fundamental reason of seeing eigenvalues distributed on the unit circle in the deep Koopman-layered model? Why EDMD/KDMD failed? Since you are benchmarking against DMD methods on i) non-autonomous system (transient dynamics) and the goal is to ii) find stable eigenvalues, I strongly recommend (NOT required) you to compare against more relevant DMD variants below:
  - "Multi-Resolution Dynamic Mode Decomposition" by J Kutz et al.
  - "Forward-Backward Extended DMD with an Asymptotic Stability Constraint" by Forbes et al.

---

> ### Author Response · Authors · 2024-11-19
>
> Thank you very much for your constructive comments. We addressed you comments and questions and revised the paper. We summarize the answers below.
>
> **[W1] About the statement “For existing Neural ODE-based models, we use numerical methods, such as Runge-Kutta methods, to solve the ODE (Chen et al., 2018)”**
> We are sorry for the confusion and thank you for pointing out.
> We replaced the sentence with
> "For existing Neural ODE-based models, we solve ODEs for the forward computation and solve adjoint equations for backward computation (Chen et al., 2018)".
> In addition, we replaced the last sentence with "our framework provides numerical linear algebraic way to *solve* Neural ODE-based models".
>
> **[W2] Notations and readability**
> Thank you for pointing out.
> We added the definition of $x_k$ in line 161.
> We also omit the sub- and super-script of $j$ in Section 3.2 and added the explanation "For the remaining part of this section, we omit the subscript $j$ for simplicity.
> However, in practice, the approximation is computed for the generator $L_j$ for each layer $j=1,\ldots,J$."
>
> **[W3] Additional references**
> We added the reference to section 7.
> They are deeply related to this paper.
> Thank you for the references.
>
> **[Q1] Necessity of more than one layer even for the autonomous systems**
> This is an effect of the approximation of the generator.
> If we can use the true Koopman generator, then we only need one layer for autonomous systems.
> However, since we approximated the generator using matrices, we may need more than one layer.
> We added the above explanation right after the sentence.
>
> **[Q2] Choice of $J$ (number of layers) and $v$ (fixed nonlinear function in the proposed model)**
> Regarding the choice of $J$, heuristically, we can use validation data to determine an optimal number of layers.
> For example, we begin by one layer and compute the validation loss.
> Then, we set two layers and compute the validation loss, and continue with more layers.
> We can set the number of layers as the number that achieves the minimal validation loss.
> Another way is to set a sufficiently large number of layers and train the model with the validation data.
> As we discussed in Section 6.2, we can add a regularization term to the loss function so that the Koopman layers next to each other become close.
> After the training, if there are Koopman layers next to each other and sufficiently close, then we can regard them as one Koopman layer and determine an optimal number of layers.
> We added Appendix G for the above explanation.
>
> Regarding the choice of $v$, as you pointed out, we chose $v$ so that it is suitable for the Fourier functions.
> Heuristically, we can parameterize $v$ and choose it using validation data or learn it simultaneously with the learnable parameters in the proposed model with training data.
> More deeper investigation is left to future work.
>
> **[Q3] Fundamental reason of seeing eigenvalues distributed on the unit circle in the deep Koopman-layered model and comparison against more relevant DMD variants**
> We guess that since EDMD and KDMD are for autonomous systems and we applied them individually for each time-window, they tried to interpret the underlined dynamical system as an autonomous system, and fail to capture nonautonomous behavior of the system.
> Based on other reviewers' comments, we added numerical results regarding the damping oscillator with external force (Section 6.3.2).
> We conducted experiments with the Koopman-based approach with learned dictionary functions.
> We considered the following two settings.
>
> 1. Learn a set of dictionary functions to construct the representation space of five Koopman generators. (learning a common set of dictionary functions is also considered in [1,2])
> 2. Learn five sets of dictionary functions each of which is for each Koopman generator.
>
> We used a 3-layered ReLU neural network to learn the dictionary functions.
> The result is illustrated in Appendix E.
> We cannot capture the transition of the distribution of the eigenvalues through $j=1,\ldots,5$ even though we learned the dictionary functions.
> We can also see that there are some eigenvalues equally spaced on the unit circle.
> This behavior is typical for autonomous systems with a constant frequency.
> Since the dynamical system is nonautonomous and the frequency of the system changes over time, the above behavior is not suitable for this example.
> We conclude that the proposed method can capture the transition of the dynamics more properly than DMD-based method regardless of whether the dictionary functions are fixed or not.
>
> We also implemented the forward-backward extended DMD (the second reference) with learned dictionary functions, based on your comment (The first one also seems to be deeply related to the proposed method, but since it is mainly for the standard DMD without Koopman operators, we focused more on the second reference).
> The result is also shown in Appendix E, and it is similar to the above two settings.

---

> > ### Author Response · Authors · 2024-11-19
> >
> > References:
> > [1] Rui Wang, Yihe Dong, Sercan O. Arik, and Rose Yu, "Koopman Neural Operator Forecaster for Time-series with Temporal Distributional Shifts", ICLR2023.
> > [2] Yong Liu, Chenyu Li, Jianmin Wang, and Mingsheng Long, "Koopa: Learning non-stationary time series dynamics with Koopman predictors", NeurIPS 2023.

---

> > > ### Comment · Reviewer_q7PE · 2024-11-26
> > >
> > > I appreciate the response from the authors, and my concerns are properly addressed given these clarifications. However, given the scope of the innovation (as pointed out by other reviewers as well), I'd still keep it as a regular acceptance.

---

> > > > ### Author Response · Authors · 2024-11-26
> > > >
> > > > Thank you very much for your response. If you have any additional comments, questions, please let us know.

---

### Official Review · Reviewer_naMD · 2024-11-03

**Soundness:** 2
**Presentation:** 2
**Contribution:** 2
**Rating:** 5
**Confidence:** 5

**Summary:**

The authors introduce a concatenation of Koopman operator approximations ("Deep Koopman model") defined on a Fourier approximation of the L^2 space on an n-torus, for the approximation of nonlinear, nonautonomous dynamical systems.
They propose to learn the generator in each of the Koopman operators (i.e., in each "layer" of the model) as Toeplitz matrices, and then map the learned generators to the corresponding operator using matrix exponentials.
The model is analyzed regarding universal approximation and generalization, and compared to KDMD and EDMD on several numerical examples on the torus.

**Strengths:**

Constructing models for non-autonomous, non-linear dynamical systems is a very important and very challenging problem. On the n-torus, Fourier series provide a very efficient and useful dictioanry to approximate functions in L2, so the setting the authors chose to present their Koopman approximation is reasonable. The theoretical contributions cover most of the important, first aspects that should be treated when introducing a new model, and seem to be proven appropriately. The numerical examples are chosen well, and illustrate the efficacy of the new method. The connection to neural ODEs (or rather, approximation of ODE vector fields using neural networks) is interesting and should be explored further.

**Weaknesses:**

The authors chose an extremely specific setting (systems on n-tori) and can therefore avoid the most challenging question in Koopman operator approximation - how to choose the proper dictionary for a given problem? The challenging aspect in their setting thus only comes from the fact that the chosen systems are non-autonomous, not that they are non-linear. Unfortunately, the authors do not discuss any existing methods for non-autonomous systems or compare their method in the numerical setting against them. In fact, the "deep Koopman model" proposed by the authors may not even be appropriate for general non-autonomous systems, because the number of internal "layers" may need to go to infinity for such systems, e.g. if the time-dependent part is not periodic (but, e.g., linear: $\dot{x}= -x+t$). I list those concerns in more detail below.

 * One of the most challenging aspects of numerical Koopman operator approximation is the choice of function space and corresponding truncation. The authors only work on the n-torus, for which the Fourier basis is a good choice - but this also means that they do not discuss (at all) the issues arising when the base space is not an n-torus (which, I would argue, is a very common case in practice).

 * The authors should incorporate and discuss existing literature on learning dictionary functions and switched systems:
   - Li, Qianxiao, Felix Dietrich, Erik M. Bollt, and Ioannis G. Kevrekidis. “Extended Dynamic Mode Decomposition with Dictionary Learning: A Data-Driven Adaptive Spectral Decomposition of the Koopman Operator.” Chaos: An Interdisciplinary Journal of Nonlinear Science 27, no. 10 (October 2017): 103111. https://doi.org/10.1063/1.4993854.
   - Peitz, Sebastian, and Stefan Klus. “Koopman Operator-Based Model Reduction for Switched-System Control of PDEs.” Automatica 106 (August 2019): 184–91. https://doi.org/10.1016/j.automatica.2019.05.016.

 * Experiment 6.2.1 is not using proper dictionary functions for KDMD and EDMD, and hence the numerical results are unfavourable for these methods compared to the presented method. The authors should either learn Koopman operators separately for each subset, or use time-dependent dictionary functions for EDMD - the true model is periodic with period $2\pi$, so it should be easily possible to properly encode the dynamics even with classical EDMD (and time-dependent dictionary).

**Questions:**

* I do not understand the sentence in the introduction: "In addition, since each Koopman operator for a time window is estimated individually in these frameworks, we cannot take the information of other Koopman operators into account.". A single system only has a single Koopman operator (family, if one considers one operator per time t). What do the authors mean by "other Koopman operators"? The family?

 * For proposition 5.1: is it not possible to use simple Monte-Carlo arguments to show that the variance of $h(x,y)$ goes to zero with $1/\sqrt{S}$ (i.e., $1/S \sum_i(h(x_i,y_i)$) converges to $E[h(x,y)]$ with the Monte-Carlo rate of $1/\sqrt{S})$? The current bound involves operator norms which may be unbounded, correct?

 * l478 "our framework provides numerical linear algebraic way to train Neural ODE-based models" which part of the "deep Koopman" model represents the neural network from the Neural ODE? The neural ODE framework was introduced by Duvenaud and coauthors (Chen et al. (2018)) to mitigate the memory consumption of back-propagation through classical solvers, they propose instead to use the adjoint method. Where does the memory footprint come in with the new method? "Finding the vector field of an ODE" is not the novelty of Chen et al. (2018), it is finding it *efficiently* (without a lot of memory cost).

 * How many internal layers would one need to approxiamte the simple non-autonomous system "\dot{x} = -x+t"?

---

> ### Author Response · Authors · 2024-11-19
>
> Thank you very much for your constructive comments. We addressed you comments and questions and revised the paper. We summarize the answers below.
>
> **[W1] Issues arising when the base space is not an n-torus**
> We do not think the analysis in the n-torus is a specific setting.
> In many practical cases, we are interested in  dynamics in a bounded domain $\Omega$ in $\mathbb{R}^d$.
> For example, dynamics in a space around a certain object (e.g., heat source).
> In more physical settings, we are often interested in the dynamics around a limit cycle or an equilibrium point.
> Let $S^{d-1}$ be the $(d-1)$-dimensional sphere.
> If $\Omega$ is diffeomorphic to $S^{d-1}$, then we can construct a dynamical system $\check{f}_j$ on the torus $\mathbb{T}^d$ that satisfies $\check{f}_j(x)=\tilde{f}_j(x)$ for $x\in S^{d-1}$, where $\tilde{f}_j$ is the equivalent dynamical system on $S^{d-1}$ with $f_j$.
> For more details, please see Appendix B.
> We added Remark 3.2 for the above argument.
>
> We agree that in some cases, it would be reasonable to consider $S^d$ directly or other types of domains (e.g. $\mathbb{R}^d$).
> In these cases, we can use the spherical harmonics or wavelet basis functions.
> However, the theoretical analysis in these cases is challenging and is left to future work.
>
> **[W2] Discussion of existing literature on learning dictionary functions and switched systems**
> Thank you for the references.
> We added the references.
> Regarding the first reference, we added it to Section 7.2 since in our understandings, it is mainly for autonomous systems.
> Regarding the second reference, in our understandings, each Koopman operator for each time window is estimated individually.
> We added it to Section 1.
>
> We emphasize that an advantage of the proposed method over these approaches is that it has both theoretical solidness and flexibility.
> By fixing the dictionary functions, we can derive a strong theoretical guarantee; we can represent any function using the proposed model.
> At the same time, we can learn multiple Koopman operators simultaneously, which enables us to extract the transition of properties of dynamical systems along time.
>
> **[W3] Dictionary functions for EDMD and KDMD in the experiment**
> We used the same Fourier dictionary functions as the proposed deep Koopman-layered model for EDMD.
> Also, we applied the principal component analysis to select better dictionary functions for KDMD.
> Thus, we believe they are proper dictionary functions for the comparison with the proposed model.
> However, we agree that comparing the proposed method only to the method with fixed dictionary functions was not enough.
> Thus, we conducted additional experiments with the Koopman-based approach with learned dictionary functions.
> We considered the following two settings.
>
> 1. Learn a set of dictionary functions to construct the representation space of five Koopman generators. (learning a common set of dictionary functions is also considered in [1,2])
> 2. Learn five sets of dictionary functions each of which is for each Koopman generator.
>
> We used a 3-layered ReLU neural network to learn the dictionary functions.
> The result is illustrated in Appendix E.
> We cannot capture the transition of the distribution of the eigenvalues through $j=1,\ldots,5$ even though we learned the dictionary functions.
> We can also see that there are some eigenvalues equally spaced on the unit circle.
> This behavior is typical for autonomous systems with a constant frequency.
> Since the dynamical system is nonautonomous and the frequency of the system changes over time, the above behavior is not suitable for this example.
> We conclude that the proposed method can capture the transition of the dynamics more properly than DMD-based method regardless of whether the dictionary functions are fixed or not.
>
> **[Q1] About the sentence ""In addition, since each Koopman operator for a time window is estimated individually in these frameworks, we cannot take the information of other Koopman operators into account."**
> We are sorry for the confusion.
> This is about the existing works based on EDMD for nonautonomous systems with fixed dictionary functions.
> In these methods, each Koopman operator for each time-window is estimated separately.
> We revised the sentence.
>
> [1] Rui Wang, Yihe Dong, Sercan O. Arik, and Rose Yu, "Koopman Neural Operator Forecaster for Time-series with Temporal Distributional Shifts", ICLR2023.
> [2] Yong Liu, Chenyu Li, Jianmin Wang, and Mingsheng Long, "Koopa: Learning non-stationary time series dynamics with Koopman predictors", NeurIPS 2023.

---

> > ### Author Response · Authors · 2024-11-19
> >
> > **[Q2] Monte-Carlo arguments to show that the variance of $h(x,y)$ goes to zero with $1/\sqrt{S}$**
> > Thank you for the suggestion.
> > A goal of Section 5 is to investigate the relationship between the generalization error and the learnable parameters of the proposed model and find how we can regularize the optimization for better generalization property.
> > For this purpose, we used the Rademacher complexity of the model to derive the generalization bound.
> > It was not clear for us how we can relate this purpose to the Monte-Carlo arguments.
> >
> > Proposition 5.1 implies that if the norm of the operators are bounded, then the Rademacher complexity is bounded above by using the norm of the operators.
> > In practice, by adding a regularization term according to the bound to the loss function, we can bound the right hand side of the formula in Proposition 5.1, which results in a better generalization property.
> >
> > **[Q3] About the sentence "our framework provides numerical linear algebraic way to train Neural ODE-based models" which part of the "deep Koopman"**
> > We are sorry for the confusion and thank you for pointing out.
> > We admit that the explanation was inappropriate.
> > We replaced the sentence with
> > "For existing Neural ODE-based models, we solve ODEs for the forward computation and solve adjoint equations for backward computation (Chen et al., 2018)".
> > In addition, we replaced the last sentence with "our framework provides numerical linear algebraic way to *solve* Neural ODE-based models".
> >
> > **[Q4] Number of internal layers needed**
> > Although providing thorough discussion of determining an optimal number $J$ of layers and investigate some examples is future work, we provide examples of heuristic approaches to determining $J$.
> > Heuristically, we can use validation data to determine an optimal number of layers.
> > For example, we begin by one layer and compute the validation loss.
> > Then, we set two layers and compute the validation loss, and continue with more layers.
> > We can set the number of layers as the number that achieves the minimal validation loss.
> > Another way is to set a sufficiently large number of layers and train the model with the validation data.
> > As we discussed in Section 6.2, we can add a regularization term to the loss function so that the Koopman layers next to each other become close.
> > After the training, if there are Koopman layers next to each other and sufficiently close, then we can regard them as one Koopman layer and determine an optimal number of layers.

---

> > > ### Comment · Reviewer_naMD · 2024-11-25
> > > **Convergence and non-autonomous, non-periodic systems**
> > >
> > > I appreciate the response, the additional experiments, and the changes in the paper. I still have the same reservations as before, though, so I will not raise my score.
> > >
> > > [W1] Using a general diffeomorphism on a manifold can drastically change the type of dictionary that is "good" for that manifold. I agree with the authors that many shapes can be deformed into their setting (especially when extensions are used), but that does not mean that Fourier series are the best dictionary for all cases. With the same argument, one could say that polynomials (or neural networks, for that matter) are the best dictionary, because they also can be used to represent any continuous function. The new analysis also has typos (unless the authors want to represent the ball, not the sphere, in appendix B; it should be $\|x\|=1$, not $\|x\|\leq 1$).
> > >
> > > [W3] The additional experiments do not use the correct dictionary either. The basis functions should depend on time, which can also be learned dictionary functions. The authors could also use the "switched system" paper by Peitz et al., discretize time into a sufficiently fine sequence (say, 100 intervals over [0,2pi]) and then switch after each interval.
> > >
> > > [Q2] I was not wondering about the norm of the approximate operators (which is bounded because they are finite) - my concern was related to the true operator, which can be unbounded in the function space. In that case, it may be that the norm of the finite operators also will grow too quickly (even though it is always bounded for each fixed epsilon) to be of any use. The main problem is that the theoretical result uses an existence argument, and does not include a rate that relates N with the true operator - which, of course, is much harder to obtain theoretically. If the error is only reduced by $1/\sqrt{S}$, but the norm $\|\exp(L)\|$ is arbitrarily large (depending on epsilon), then the number of samples required is prohibitively large.
> > >
> > > [Q4] Apologies for my brief comment before. I was actually curious about the concrete system I provided, not a general algorithm (or heuristic) to determine the number of layers. The system $\dot{x}=-x+t$ is special because its solution is growing steadily in time, and not periodic. The heuristic that the authors provide now would just keep adding layers forever, because the validation error will never shrink (unless the trajectories are cut off).

---

> > > > ### Author Response · Authors · 2024-11-26
> > > >
> > > > Thank you very much for the additional comments and clarifications. Here are our response for each comment.
> > > >
> > > > **[W1]** We have the following two important features of the Fourier functions that makes the analysis based on Koopman operators:
> > > > - They induce a universality of the model with any fixed nonlinear function $v\in L^2_0(\mathbb{T}^d)$ (corresponding to the observable). The representation of any continuous function do not mean the universality of the model. An important fact is that applying the Fourier functions, we can represent the Koopman generator with Toeplitz matrices, which induces the universality of the model since any matrix is represented by the product of Toeplitz matrices. If we apply polynomials, we cannot have this type of property.
> > > > - They makes the algorithm computationally efficient. Applying the fast Fourier transform, we can compute the product of an n by n Toeplitz matrix and a vector with the computational cost of $O(n\log n)$. Thus, even for the resulting matrices are not sparse, we can compute the output of the model efficiently. If we apply polynomials, we cannot have this type of property. We are sorry that we did not include this second point in the original version of our paper. In the revised version, based on the discussion with another reviewer, we added this point before Remark 3.4.
> > > > Thank you also for pointing out the typos. We corrected them.
> > > >
> > > > **[W3]** For the second setting of the additional experiment, we used a different dictionary function for each time step. As you pointed out, we can also use finer time steps for learning dictionary functions. However, we do not think this setting is fair. In that case, we need to compare the above setting with the proposed deep Koopman-layered model with more layers. The time step for each method should be the same for the fair comparison, and we showed that under the same time step setting, the proposed method captured the dynamics more properly than the EDMD with learned dictionary functions.
> > > >
> > > > **[Q2]** This is an important point, and thank you for pointing out. Since we are considering the Koopman operator on $L^2$ space, the Koopman operators are bounded if the flow $g(t,\cdot)$ is invertible and the Jacobian $Jg_t^{-1}$ of $g_t^{-1}$ is bounded for any $t$. Here, we denote $g_t=g(t,\cdot)$. Indeed, we have
> > > > $$ \Vert K^th\Vert^2=\int_{\mathbb{T}^d}\vert h(g(t,x))\vert^2 dx = \int_{\mathbb{T}^d}\vert h(x)\vert^2 \vert \operatorname{det}Jg_t^{-1}(x)\vert dx \le \Vert h\Vert^2 \sup_{x\in \mathbb{T}^d}\vert\operatorname{det}Jg_t^{-1}(x)\vert.$$
> > > > Thus, under the setting of invertible and bounded-derivative flows, we do not need to have a concern about the boundedness of the Koopman operators.
> > > > We clarified this assumption in Section 2.3 so that readers can understand what type of systems we are interested in. We also added the comment on the boundedness of the Koopman operators in Remark 5.4 and Appendix H.
> > > >
> > > > **[Q4]** Thank you for the clarification. As we discussed in [W1], we are focusing on systems on bounded domains. Since the system $\dot{x}=-x+t$ is not bounded, dealing with this system is beyond the scope of this paper. We would like to emphasize that this work is the first step to construct a Koopman-based method with both the theoretical guarantee and the flexibility. Expanding the method to systems on an unbounded domain is an interesting direction of future work.

---

> > > > > ### Comment · Reviewer_naMD · 2024-11-26
> > > > >
> > > > > I appreciate the additional explanations. I will raise my score slightly, because the explanations in the paper help to clarify the methodology. W3, Q3 and Q4 are adequately addressed. There are still typos in the manuscript (e.g. l215, "fast Fourier transform", not "first"), and the general text is also hard to understand for non-experts, so I do not recommend publication. Separately:
> > > > >
> > > > > [W1] The arguments of authors go in the direction "Fourier series have nice properties", which is of course the main reason to use them. My criticism was more directed in the other direction: for cases where Fourier series are not ideal (embedded submanifolds, manifolds not diffeomorphic to tori) the benefits would not apply. The authors do not need to solve the entire problem regarding "choice of dictionary", but in the current text it is stated as if this was not an issue. Focusing on systems on tori would actually strengthen the presentation, I believe, because it would not suggest that the method is generally applicable (the authors do not demonstrate it is generally applicable away from tori systems, either).

---

> ### Author Response · Authors · 2024-11-26
>
> Thank you very much for your response and additional comments, and thank you for updating the score.
>
> Although we focus on the fundamental case of $\mathbb{T}^d$, the analysis on $\mathbb{T}^d$ opens up methods for more general cases.
> Indeed, $\mathbb{T}^d$ is the simplest example of locally compact groups, and the Fourier functions are generalized to the irreducible representations [1].
> For a group $G$, a representation $\rho$ is a map $\rho:G\to GL(V)$ for some vector space $V$ that satisfies $\rho(x)\rho(y)=\rho(x\cdot y)$ for $x,y\in G$.
> Here, $GL(V)$ is the space of all bijective linear transformations from $V$ to $V$.
> We note that if $G$ is abelian, then $V$ is always one-dimensional.
> For example, for $\mathbb{R}^d$, the irreducible representations are $\rho\_\{\xi\}(x):=\mathrm{e}^{\mathrm{i}\xi \cdot x}$ for $x\in\mathbb{R}^d$ and $\xi\in\mathbb{R}^d$.
> As in Section 3.2, a crucial property of the Fourier function $q_n$ is the product of two Fourier functions are also a Fourier function, i.e., $q_n\cdot q_m=q_{m+n}$.
> This property is valid also for general irreducible representations since we have $\rho(x)\rho(y)=\rho(x\cdot y)$.
> Thus, we can consider a generalized version of Toeplitz matrices by using $\rho$ [2].
> Although showing the universal property of the generalized Toeplitz matrices is challenging and future work, this type of argument gives us a promising way of generalizing our framework.
> We added this argument in Remark 3.2 and Appendix B.2.
>
> We would like to emphasize that this work is the first step for constructing a method both with the theoretical guarantee and the flexibility. The torus is the most fundamental example of locally compact groups, and although there are some mathematically challenging problems, we have a clear path for the generalization of the proposed method.
>
> We also added some explanations in the main text to improve the readability and corrected the typo. The revised parts are colored in blue. We would like to note that based on other reviewers' comments, we improved Subsection 6.1 and added Appendix E to help readers to understand the algorithm well. Also, we omitted the sub- and super-script $j$ in Section 3.2 to simplify the notation.
>
> [1] William Fulton and Joe Harris. Representation Theory –A First Course–, Springer, 2004.
> [2] Marc A. Rieffel. Matrix algebras converge to the sphere for quantum Gromov–Hausdorff distance. Memoirs of the American Mathematical Society, 168:67-91, 2004.

---

### Official Review · Reviewer_V5co · 2024-11-03

**Soundness:** 2
**Presentation:** 1
**Contribution:** 2
**Rating:** 5
**Confidence:** 2

**Summary:**

The authors present an approach for learning Koopman representations of time-series datasets. The main idea is to transform the data into a new space using fourier functions where the dynamics can be approximated by a switching linear dynamical system. They also propose to use Krylov subspace methods to approximate the matrix exponentials which define the time evolution of the linear system.

**Strengths:**

- Transforming the original space into the space of observables using fixed basis functions (as opposed to learning the space of observables like in Lusch 2017) allows the authors to develop a strong theoretical backing for their approach including showing universality and a generalization bound. I think the theoretical foundation of the work is its strongest feature.
- A nice feature of the approach is that one can analyze the eigenvalues of the trained model to interpret the underlying system (for example whether it is measure preserving or not).
- The idea of using Krylov subspace methods to approximate the matrix exponential matrix product in this context is novel as far as I'm aware.

**Weaknesses:**

- Overall, I found it quite challenging to understand the details of your proposed learning algorithm and I am not confident that I would be able to reproduce your approach using your paper alone (that said, I appreciate the authors providing their code in the supplementary materials). It would have been helpful for me if you had included a section which provides a step-by-step summary or an algorithm block showing how your approach works for a given time-series dataset.
- I think the claims you make in S7 (that your approach provides an alternative to neural ODE-based models or other deep Koopman based approaches) is not well-supported by your numerical studies. To put these claims on solid footing you need to include numerical studies showing your approach can provide equivalent performance on some common time-series benchmarks; for example see the numerical studies [1].
- It wasn't clear to me how your approach can be used in time-series forecasting (i.e. making predictions for $t>t_J$.

[1] Wang, R., Dong, Y., Arik, S. Ö., & Yu, R. (2022). Koopman neural forecaster for time series with temporal distribution shifts. arXiv preprint arXiv:2210.03675.

Overall because the presentation of your approach was challenging to parse and because your numerical studies did not do a good job supporting some of your claims, I feel that I cannot recommend acceptance of your work at this time.

*Minor comments which will not impact my decision*
- I noticed a couple of spelling mistakes -- please run your draft through a spell checker for the next submission.

**Questions:**

- In my understanding of your work, you learn an approximation to a time-series dataset over the time-window $t \in [t_1, t_J]$. In most time-series forecasting problems, one goal is to generate forecasts beyond the time-window in which the observations were collected. With your approach how do you generate forecasts for $t \geq t_J$?
- It wasn't clear to me under which conditions the Toeplitz matrix will be sparse in practice. Can you discuss in which cases this will be true? How will your approach scale in cases where you cannot make this assumption?
- My understanding is that you haven't placed requirements on your Fourier functions that they be invertible. In this case, is it possible to reconstruct the original space once you've made a forecast in the observable space?

---

> ### Author Response · Authors · 2024-11-19
>
> Thank you very much for your constructive comments. We addressed you comments and questions and revised the paper. We summarize the answers below.
>
> **[W1] Readability to understand the details of the proposed learning algorithm**
> We moved the paragraph that explains the details of the training the deep Koopman-layered model in Section 4 to the beginning of Section 6 as a new section 6.1.
> We also added more detailed explanation of how to compute the output of the model so that readers can understand the details of the proposed learning algorithm.
> Moreover, we added the pseudocode and more details in Appendix D.
> We hope the revised version is helpful for readers to understand the details of the algorithm and reproduce it.
>
> **[W2] Numerical studies to show the proposed approach can provide equivalent performance on some common time-series benchmarks**
> An advantage of the Koopman-based approaches over neural ODE-based approaches is that we can extract the information of the transition of the dynamics by computing the eigenvalues of the approximated Koopman operators.
> Therefore, we compared the proposed method to other Koopman-based approaches in the sense that how we can extract the information of the transition of the dynamics.
> We admit that comparing the proposed method only to the method with fixed dictionary functions was not enough.
> Thus, we conducted additional experiments with the Koopman-based approach with learned dictionary functions.
> We considered the following two settings.
>
> 1. Learn a set of dictionary functions to construct the representation space of five Koopman generators. (learning a common set of dictionary functions is also considered in [1,2])
> 2. Learn five sets of dictionary functions each of which is for each Koopman generator.
>
> We used a 3-layered ReLU neural network to learn the dictionary functions.
> The result is illustrated in Appendix E.
> We cannot capture the transition of the distribution of the eigenvalues through $j=1,\ldots,5$ even though we learned the dictionary functions.
> We can also see that there are some eigenvalues equally spaced on the unit circle.
> This behavior is typical for autonomous systems with a constant frequency.
> Since the dynamical system is nonautonomous and the frequency of the system changes over time, the above behavior is not suitable for this example.
> We conclude that the proposed method can capture the transition of the dynamics more properly than DMD-based method regardless of whether the dictionary functions are fixed or not.
>
> **[W3, Q1] Application to time-series forecasting for $t>t_J$**
> As we also insist above, an advantage of the proposed method is that we can extract the information of the transition of the dynamics by computing the eigenvalues of the approximated Koopman operators.
> We can also forecast the time-series starting with a different initial value for $t\le t_J$.
> Our main goal in this paper is not the time-series forecasting for $t>t_J$.
> However, as you pointed out, applying the proposed method to time-series forecasting is also interesting.
> Applying the idea of [1,2], we can decompose the Koopman operators into time-invariant and time-variant parts.
> By extracting time-invariant features of the dynamics using the approximated Koopman operators (e.g., time-invariant eigenvectors or singular vectors), we can combine it with time-variant Koopman operators constructed by local time-series to construct the forecast.
>
> More precisely, we can decompose the Koopman operator $K^t$ for time $t$ as
> $K^t=K_{inv}+K^t_{var}$, where $K_{inv}=\sum_{i=1}^n\sigma_iv_iu_i^*$ and $K_{var}=\sum_{i=1}^m\tilde{\sigma}_i\tilde{v}_i\tilde{u}_i^*$, $\sigma_i$, $v_i$, $u_i$ are time-invariant singular values and the corresponding singular vectors of the approximated Koopman operators for $j=1,\ldots, J$, $\tilde{v}_i$ are the singular vectors of the local Koopman operator that is orthogonal to $v_i$, and $\tilde{\sigma}_i$ and $\tilde{u}_i$ are singular values and singular vectors corresponding to $\tilde{v}_i$.
> Since we can use the time-invariant property of $t\le t_J$, we can forecast time-series well even for $t>t_J$.
> We added Appendix F for explaining the above point.
>
> [1] Rui Wang, Yihe Dong, Sercan O. Arik, and Rose Yu, "Koopman Neural Operator Forecaster for Time-series with Temporal Distributional Shifts", ICLR2023.
> [2] Yong Liu, Chenyu Li, Jianmin Wang, and Mingsheng Long, "Koopa: Learning non-stationary time series dynamics with Koopman predictors", NeurIPS 2023.

---

> > ### Author Response · Authors · 2024-11-19
> >
> > **[Q2] Conditions for the Toeplitz matrix being sparce**
> > The sparseness of the Toeplitz matrix is characterized by $M_r^j$, the index set for representing the dynamical system $f_j$ with the Fourier functions.
> > Thus, if $f_j$ has few Fourier components, then the corresponding Toeplitz matrix is sparse.
> > If $f_j$ has a lot of Fourier components, the Toeplitz matrix is not sparse.
> > In this case, we can approximate the Fourier components with small coefficients by 0 to make the Toeplitz matrix sparse and for efficient computations.
> >
> > **[Q3] Invertibility of the Fourier functions**
> > The Fourier functions are not invertible.
> > However, we do not need the inverse of the Fourier functions.
> > As we explained in Remark 3.1, if we construct a model on $\mathbb{T}^{d+1}$ and set $\tilde{v}((x_1,\ldots,x_d),k/d)=x_k$, then we can reconstruct the forecast in the original space.

---

> > ### Comment · Reviewer_V5co · 2024-11-26
> >
> > Thank you for making the edits to Section 6 as well as adding more details of your algorithm in Appendix D. This has made the paper much more clear to me. I also appreciate you taking the time to discuss the application of your approach to time-series forecasting beyond the window $t>t_J$.
> >
> > I still think that your claim that your approach provides an alternative to neural ODE-based approaches is not well supported by the numerical studies you have already completed. I have increased my score to a 5.

---

> > > ### Author Response · Authors · 2024-11-27
> > >
> > > Thank you very much for your response and for updating the score.
> > >
> > > To address your concern about the comparison to neural ODE-based models, we conducted an additional experiment. We compared the performance of the deep Koopman-layered model considered in Subsection 6.2 to a nural ODE-based model with almost the same number of learnable parameters as the deep Koopman-layered model. We solved the same task as that in Subsection 6.2. Please see Appendix E.1 for more details and the result. We can see that the deep Koopman-layered model outperforms the neural ODE, which shows that the Koopman-layered model has a potential power of being an alternative to neural ODE-based approaches.
> > >
> > > We would like to emphasize that an advantage of the Koopman-based approaches over neural ODE-based approaches is that we can extract the information of the transition of the dynamics by computing the eigenvalues of the approximated Koopman operators. In addition to this advantage, we also confirmed that the proposed deep Koopman-layered model is a promissing approach that can also be applied to tasks for neural ODE-based approaches.

---

> > > > ### Author Response · Authors · 2024-11-28
> > > >
> > > > Dear Reviewer V5co,
> > > >
> > > > Apologies for our continuous messages. Since the deadline of revising the paper is approaching, we updated the paper by adding one more numerical result in Appendix E.1 and Figure 5. We showed even if we use two time steps for training the neural ODE model, the Koopman layered model outperforms the neural ODE model. We believe that the result also shows that the Koopman-layered model has a potential power of being an alternative to neural ODE-based approaches.
> > > >
> > > > We hope the additional numerical results address your concerns. Thank you again for your time and valuable comments.

---

### Official Review · Reviewer_dHBH · 2024-11-08

**Soundness:** 3
**Presentation:** 2
**Contribution:** 2
**Rating:** 5
**Confidence:** 4

**Summary:**

The paper presents a novel deep Koopman-layered framework for modeling dynamical systems, particularly suited for nonautonomous time-series data. This approach integrates Koopman operator theory with Fourier functions and learnable Toeplitz matrices, enabling the simultaneous estimation of multiple Koopman operators to capture temporal dynamics. The model is theoretically robust and flexible, leveraging the universal and reproducing properties of Toeplitz matrices, which enhance its generalization capabilities. Efficient training is achieved through Krylov subspace methods.

**Strengths:**

1. **Originality**
The paper introduces an innovative integration of deep learning techniques with Koopman operator theory. By proposing deep Koopman-layered models that utilize learnable Toeplitz matrices and Fourier functions, the authors offer a fresh perspective for analyzing nonautonomous systems, which are often challenging for traditional methods. The approach of simultaneously estimating multiple Koopman operators is a notable contribution that broadens the model's applicability.

2. **Quality**
The authors provide a solid mathematical framework that substantiates their claims regarding the universality and generalization properties of the proposed model.

3. **Clarity**
The paper is well-structured and almost clearly written. The authors effectively convey the motivations behind their approach and highlight the significance of Koopman operator theory.

4. **Significance**
The significance of the paper lies in its potential impact on the fields of dynamical systems and machine learning. By bridging these two areas, the proposed deep Koopman-layered models could advance the analysis of time-series data from nonautonomous dynamical systems.

**Weaknesses:**

1. To my understanding, the choice of the Fourier basis in the proposed deep Koopman-layered model is motivated by its universality in function representation, desirable theoretical properties for analyzing Koopman operators, flexibility in learning multiple operators simultaneously, and compatibility with efficient Krylov subspace methods for low computational cost. However, it is still unclear:

- What specific properties of the Fourier basis make it particularly suitable for capturing the dynamics of nonautonomous systems compared to wavelet bases or polynomials?

- How does the use of the Fourier basis influence the convergence rates of the learning algorithms employed in the deep Koopman-layered model?

- Can the model's performance be enhanced by incorporating additional basis functions alongside the Fourier basis, and if so, how would this be implemented?

- How does the choice of the Fourier basis compare to other potential basis functions in terms of computational efficiency and accuracy when modeling complex nonautonomous dynamical systems?

2. The theoretical and numerical results in the paper are restricted to the torus. Although this has several advantages theoretically, it may limit the applicability of the theoretical results for many cases, especially for real-world scenarios. It would be great to know how the insights gained from the toroidal analysis may provide a foundational understanding to explore and develop models for more general cases, ultimately enhancing the applicability of the theoretical results to real-world scenarios.

3. There are no experiments for chaotic dynamics. So, what happens if the training dynamics are more complex, such as in the case of (autonomous) chaotic dynamics with a **mixed Koopman spectrum**?

4. The sentence "By computing the eigenvalues of Koopman operators, we can understand the long-term behavior of the undelined dynamical systems" in the Introduction, line 29, may be better if mentioned specifically for systems with a discrete Koopman spectrum, but not in general (for example, not for systems that exhibit a continuous spectrum rather than a purely discrete set of eigenvalues).

**Questions:**

I) What are some potential real-world applications of the deep Koopman-layered model?

II)  The paper discusses the concept of multiple Koopman layers. What criteria should be used to determine the **optimal number of layers** in the model? Could you provide guidelines or heuristics for selecting the number of layers for different types of problems or datasets?

Please also see "Weaknesses".

**Details Of Ethics Concerns:**

There is no Ethics Concerns.

---

> ### Author Response · Authors · 2024-11-19
>
> Thank you very much for your constructive comments. We addressed you comments and questions and revised the paper. We summarize the answers below.
>
> **[W1-1] Specific properties of the Fourier basis making it particularly suitable for capturing the dynamics of nonautonomous systems compared to wavelet bases or polynomials**
> The following two properties of Fourier functions $q_n(x)=\mathrm{e}^{\mathrm{i}n \cdot x}$ are suitable for the deep structure of the Koopman-layered model;
> 1) They form an orthonormal basis of $L^2(\mathbb{T}^d)$ and the RKHS on $\mathbb{T}^d$ considered in Section 5.
> 2) $\frac{d q_n}{dx}=\mathrm{i}nq_n$ and $q_n\cdot q_m=q_{n+m}$ (for simplicity, we consider the case of $d=1$, but same properties are valid for $d\ge 2$).
>
> The first property enables us to construct the model in a dense subspace of $L^2(\mathbb{T}^d)$ and the RKHS and to derive generalization bound (since they are also an orthonormal basis of an RKHS).
> By the second property, if a Koopman generator acts on a Fourier function, the resulting function is also represented by the weighted sum of the Fourier functions.
> This feature is useful for the deep structure of the Koopman-layered model.
> In addition, the second property enables us to approximate Koopman operators by Toeplitz and diagonal matrices, which have nice properties such as the universality and the computational efficiency (please see below for further details).
> By virtue of the above two properties of the Fourier functions, we have the theoretical advantages (universality and generalization bound) and the computational advantage.
>
> **[W1-2] Influence of the use of the Fourier basis on the convergence rates of the learning algorithms**
> In the same manner as Theorem 4.1, we can show that we can represent any function in $V_N=\operatorname{Span}\\{q_n\,\mid\,n\in N\\}$ exactly using the deep Koopman-layered model, where $N$ is the index set characterizes the approximation space of the Koopman operators.
> Thus, if the decay rate of the Fourier transform of the target function $h$ is $\alpha$, then the convergence rate with respect to $N$ is $O(({1-\alpha^2})^{-d/2})$.
> We added Remark 4.4 and Appendix C regarding the above argument. Please see Appendix C for more details.
>
> **[W1-3]  Performance enhancement by incorporating additional basis functions alongside the Fourier basis**
> Thank you for the suggestion.
> Formally, we can replace some of the Fourier basis functions with other functions.
> However, in that case, the approximated Koopman operators have more complicated structures and it is challenging to analyze it theoretically.
> Moreover, the computational cost may become expensive since the operators are not represented by Toeplitz matrices any more.
> On the other hand, by adding other functions than the Fourier functions, the convergence rate may become better.
> This paper is the first step to propose a theoretically guaranteed method for analyzing nonautonomous dynamical systems using Koopman operators.
> Investigating more sophisticated way of choosing basis functions is future work.
>
> **[W1-4] Comparison of Fourier basis to other potential basis functions in terms of computational efficiency and accuracy**
> In terms of the computational efficiency, the product of an $n$ by $n$ Toeplitz matrix and a vector is efficiently computed with the cost of $O(n\log n)$ by using the fast Fourier transform (FFT).
> As we discussed before Remark 3.4, if we set the index set $M_r$ much smaller than $N$, then the Toeplitz matrix is sparse.
> However, even if the Toeplitz matrices are dense, the computational cost of one iteration of the Krylov subspace method for computing $\mathrm{e}^{\mathbf{L}}v$ for a vector $v$ is $O(\vert N\vert \log\vert N\vert)$.
> If we use other basis functions such as wavelet and polynomial functions, then the approximation of a Koopman generator do not have a structure, and we have to deal with it as a standard dense matrix.
> Then, the computational cost is $O(\vert N\vert^2)$.
> Thank you for pointing out, and we added the explanation before Remark 3.4.
> In terms of the accuracy, as we insisted in the answer of the second question (about the convergence rate), we can represent any function in $\operatorname{Span}\\{q_n\,\mid\,n\in N\\}$ exactly using the deep Koopman-layered model.

---

> ### Author Response · Authors · 2024-11-19
>
> **[W2] Restriction of the theoretical and numerical results to the torus**
> We do not think the analysis in the torus limits the applicability of the proposed model for many cases.
> In many practical cases, we are interested in  dynamics in a bounded domain $\Omega$ in $\mathbb{R}^d$.
> For example, dynamics in a space around a certain object (e.g., heat source).
> In more physical settings, we are often interested in the dynamics around a limit cycle or an equilibrium point.
> Let $B_d$ be the unit ball in $\mathbb{T}^d$.
> If $\Omega$ is diffeomorphic to $B_d$, then we can construct a dynamical system $\check{f}_j$ on the torus $\mathbb{T}^d$ that satisfies $\check{f}_j(x)=\tilde{f}_j(x)$ for $x\in B_d$, where $\tilde{f}_j$ is the equivalent dynamical system on $B_d$ with $f_j$.
> For more details, please see Appendix B.
> We added Remark 3.2 for the above argument.
>
> We agree that in some cases, it would be reasonable to consider other types of domains (e.g. $\mathbb{R}^d$ and $S^{d}$).
> In these cases, we can use the spherical harmonics or wavelet basis functions, as you pointed out.
> However, the theoretical analysis in these cases is challenging and is left to future work.
> (We corrected typos in this comments besed on the pointing out by Reviewer naMD.)
>
> **[W2] Experiments for chaotic dynamics**
> For chaotic dynamical systems, the Koopman operator has continuous and residual spectra.
> Thus, it is difficult to capture the behavior of the system only with the eigenvalues.
> The situation is the same for the standard DMD-based methods, and not specific for the proposed method.
> Dealing with more complex dynamics is future work.
>
> **[W4] About the sentence "By computing the eigenvalues of Koopman operators, we can understand the long-term behavior of the undelined dynamical systems"**
> Thank you for the suggestion.
> We are sorry for the confusion, and we agree that the explanation is only for the Koopman operators with discrete spectra.
> We revised the sentence.
>
> **[Q1] Potential real-world applications**
> As we discussed above, for example, we can analyze dynamics around a certain object (e.g. heat source) in $\mathbb{R}^d$ using the proposed model.
> For the example of a heat source, the amount of heat created by the source may switch or change over time.
> In addition, the environment around the source may also switch or change over time (e.g. by wind, by humans).
> In this case, we can use the proposed model to analyze the transition of the temperature around the heat source.
>
> **[Q2] Optimal number of layers**
> Heuristically, we can use validation data to determine an optimal number of layers.
> For example, we begin by one layer and compute the validation loss.
> Then, we set two layers and compute the validation loss, and continue with more layers.
> We can set the number of layers as the number that achieves the minimal validation loss.
> Another way is to set a sufficiently large number of layers and train the model with the validation data.
> As we discussed in Section 6.2, we can add a regularization term to the loss function so that the Koopman layers next to each other become close.
> After the training, if there are Koopman layers next to each other and sufficiently close, then we can regard them as one Koopman layer and determine an optimal number of layers.
> We added Appendix G for the above explanation.

---

> > ### Author Response · Authors · 2024-12-04
> >
> > Dear Reviewer dHBH,
> >
> > Reflecting the discussion with another reviewer, we added explanations that show our framework gives us a promising way of more general analysis. We can use the notion of irreducible representations to generalize the Fourier functions. We added this argument in Remark 3.2 and Appendix B.2, and please see Remark 3.2 and Appendix B.2 for more details. We believe this point is also related to your point [W2]. Thank you for your time.

---

### Author Response · Authors · 2024-11-19

We thank all the reviewers for their constructive comments and questions. We addressed them and revised the paper based on the comments. The revised parts are colored in red. The answer of each comment and quastion is posted as a comment for each reviewer.

---

### Comment · Area_Chair_xyYh · 2024-11-26

Dear Reviewers dHBH, V5co, q7PE,
If not already, could you please take a look at the authors' rebuttal? Thank you for this important service.
-AC

---

### Meta-Review · Area_Chair_xyYh · 2024-12-20

**Metareview:**

This paper proposes to compose a series of learnable Koopman operator approximations to construct "deep Koopman-layered models", for the purpose of approximating nonlinear, nonautonomous differential equations. Each Koopman operator (i.e. each "layer") has its generator represented by Toeplitz matrices so that it can represent a Fourier approximation on an n-torus. Krylov subspace methods were also employed to improve the efficiency of training. While reviewers and I feel this is an interesting approach, there were major concerns unresolved, such as applicability beyond the quasiperiodic cases, fairness in the comparison with existing approaches, and the lack of empirical demonstration on more challenging problems. Recognizing the potential of the idea, I recommend the authors take the discussions into consideration and re-submit a substantially revised version.

**Additional Comments On Reviewer Discussion:**

There were major reviewer concerns unresolved, such as applicability beyond the quasiperiodic cases, fairness in the comparison with existing approaches, and the lack of empirical demonstration on more challenging problems.

---

### Decision · Program_Chairs · 2025-01-22

Reject